# Alpha power increases spontaneously during a neurofeedback session

Jacob Maaz [1,2,3] ✉, Laurent Waroquier[4], Alexandra Dia [1,2,3], Véronique Paban[1,2] & Arnaud Rey[1,3]

Electroencephalographic neurofeedback (EEG-NF) has been proposed as a promising technique to modulate brain activity through real-time EEG-based feedback. Alpha neurofeedback in particular is believed to induce rapid self-regulation of brain rhythms, with applications in cognitive enhancement and clinical treatment. However, whether this modulation reflects specific volitional control or non-specific influences remains unresolved. In a preregistered, double-blind, sham-controlled study, we evaluated alpha upregulation in healthy participants receiving either genuine ($n = 30$) or sham ($n = 30$) EEG-NF during a single-session design. A third arm composed of a passive control group ($n = 32$) was also included to differentiate between non-specific influences related or not to the active engagement in EEG-NF. Throughout the session, alpha power increased robustly, yet independently of feedback veracity, engagement in self-regulation, or feedback update frequency. Parallel increases in theta and sensorimotor rhythms further suggest broadband non-specific modulation. Importantly, these results challenge the foundational assumption of EEG-NF: that feedback enables volitional EEG control. Instead, they point to spontaneous repetition-related processes as primary drivers, calling for a critical reassessment of neurofeedback efficacy and its underlying mechanisms.

Neural oscillations as measured by electroencephalography (EEG) have been studied as fundamental markers of brain function for almost a century[1]. Spectral densities of EEG rhythms have been extensively linked to cognitive processes such as attention, memory and perceptual binding[2]. In parallel, aberrant spectral densities have been associated with cognitive and neuropsychiatric disorders[3,4]. By linking spectral densities to both behaviour and pathology, EEG research established a framework for interventions that aim to modulate neural oscillations to influence cognitive functions and treat clinical disorders.

EEG neurofeedback (EEG-NF) is a prime example of this translational approach[5]. In a closed-loop training paradigm, individuals are provided with real-time sensory feedback contingent upon their ongoing EEG activity. Thanks to this real-time feedback, participants learn to self-regulate spectral densities of specific EEG frequency bands[6]. The resulting EEG self-regulation is presumed to significantly alter brain function, positioning EEG-NF as an effective intervention for various disorders and cognitive enhancement[7,8].

However, the fundamental assumption that behavioural/clinical improvements are specifically driven by EEG self-regulation is increasingly challenged[9]. EEG-NF protocols inherently involve numerous additional components, such as cognitive engagement in self-regulation, expectancy, motivation, and placebo effects, that may non-specifically influence EEG-NF

brain and behavioural outcomes. To disentangle true self-regulation effects from these confounds, double-blind sham-controlled designs are considered the gold standard[10]. In these designs, all participants undergo the same EEG-NF procedure but receive either genuine or sham feedback. The sham feedback is unrelated to their EEG activity, and neither participants nor experimenters know whether the feedback is genuine or sham (i.e., double-blinding). Critically, double-blind sham-controlled studies have revealed that behavioural and clinical improvements are consistently explained by non-specific influences rather than actual EEG modulation[11-13].

Besides behavioural non-specific effects, a more fundamental gap in EEG-NF research concerns the mechanisms underlying the targeted EEG modulation. EEG modulation is generally expressed by changes in targeted EEG features observed within sessions, across sessions, and between resting-state measures before and after training[14]. However, these three types of EEG changes are not systematically reported, and the mechanisms driving each of them are poorly understood, if even investigated[15]. Relatedly, a consistent lack of methodological standardisation is observed across EEG-NF protocols[8,16]. No evidence-based standards exist to guide the design of important training components such as the number of trials per session, the number and frequency of sessions, or, particularly, feedback specifications. Variations in such components may influence the learning process occurring during training, resulting in discrepancies in both the methods and

[1]Aix Marseille Univ, CNRS, CRPN, Marseille, France. [2]Institute Neuro-Marseille, NeuroSchool, Aix Marseille Univ, Marseille, France. [3]Institute of Language, Communication and the Brain, Aix Marseille Univ, Marseille, France. [4]Aix Marseille Univ, PSYCLE, Aix-en-Provence, France. ✉e-mail: jacob.maaz@univ-amu.fr

results reported. Consequently, limited knowledge hinders the proper design of EEG-NF training to maximise targeted EEG modulation[17]. If behavioural improvements are to be induced by modulating underlying EEG activity, identifying and understanding the mechanisms behind EEG modulation must become a central research goal[18].

More importantly, to operationalise EEG modulation, a key but often overlooked complication is the non-stationarity of EEG rhythms[19]. Targeted EEG spectral densities are commonly assumed to oscillate around a stable baseline, which is aimed to be up- or downregulated through training[6]. Yet, these spectral densities can drift over time independently of any feedback contingency or engagement in their self-regulation. In particular, the alpha band (8–12 Hz) upregulation has been widely used to treat clinical conditions such as post-traumatic stress and anxio-depressive disorders[20,21], as well as to enhance memory and attentional performances[22,23]. Alpha activity is considered highly susceptible to EEG-NF procedures[24], leading to effective upregulation even during single EEG-NF sessions[25–27]. However, outside of EEG-NF settings, alpha power has been shown to increase spontaneously over the course of typical one-hour visual EEG experiments without any experimental manipulation[28,29]. Alpha increases have also been reported in EEG-NF studies independently of alpha's possibility to be trained (e.g., when modulation of another feature is targeted)[22,30]. Relatedly, a recent study employed a passive visualisation protocol that replicated the temporal structure of a typical EEG-NF session[31,32]. Participants viewed a circle whose size fluctuated continuously (mirroring sham feedback) without any instruction to modulate their brain activity. Frontal, central and parietal alpha power steadily increased across trials, in line with the non-stationarity previously reported[28,29]. Overall, these findings raise urgent questions about the specificity of the EEG changes induced by alpha EEG-NF protocols.

To date, only a few exceptions have properly addressed these questions using double-blind sham-controlled designs[26,33]. Still, evidence for an actual self-regulation of alpha power remains mixed and requires further investigation[22]. Moreover, sham protocols are effectively designed to control for all non-specific influences that might affect EEG-NF outcomes[10]. Yet, these influences are not homogeneous. Some are linked to the EEG-NF procedure (e.g., participants' motivation and expectations elicited by the neuroscientific context), while others are not (e.g., fatigue and general, repetition-related effects)[14]. As a result, alpha activity may be modulated, at least in part, by both types of non-specific factors in both genuine and sham protocols.

The present study investigates the mechanisms driving alpha upregulation during a single EEG-NF session. Using a preregistered, double-blind, sham-controlled design, the primary objective was to distinguish the specific contribution of actual self-regulation from non-specific influences. Healthy young adults completed a single EEG-NF session aiming to enhance parietal alpha spectral power[25]. Participants were divided into two groups and randomly assigned to either a genuine or a sham EEG-NF session. Both groups received identical instructions and training procedures while maintaining double-blinding. Blinding success was assessed at the end of the session[14].

Importantly, a second objective was to further isolate the role of engaging in the self-regulation procedure, regardless of feedback veracity (i.e., genuine *vs.* sham), from general time-on-task effects[28,29]. Alpha trajectories in the sham group were compared to those of a third, independent group from a previous experiment[32]. This group underwent a similar sham procedure but was solely instructed to passively observe the feedback (passive visualisation task).

Furthermore, key methodological issues in the field[8,16] were addressed by manipulating the frequency of feedback update within each group. To test whether the timing of information delivery influences the learning process[34], feedback was updated at 1, 5 or 10 Hz (i.e., update in feedback display every 1000, 200 or 100 ms, respectively). Finally, to evaluate transfer self-regulation effects[14,18], participants' ability to voluntarily increase their parietal alpha power was assessed during a no-feedback transfer block. Figure 1 presents an overview of the experimental procedure.

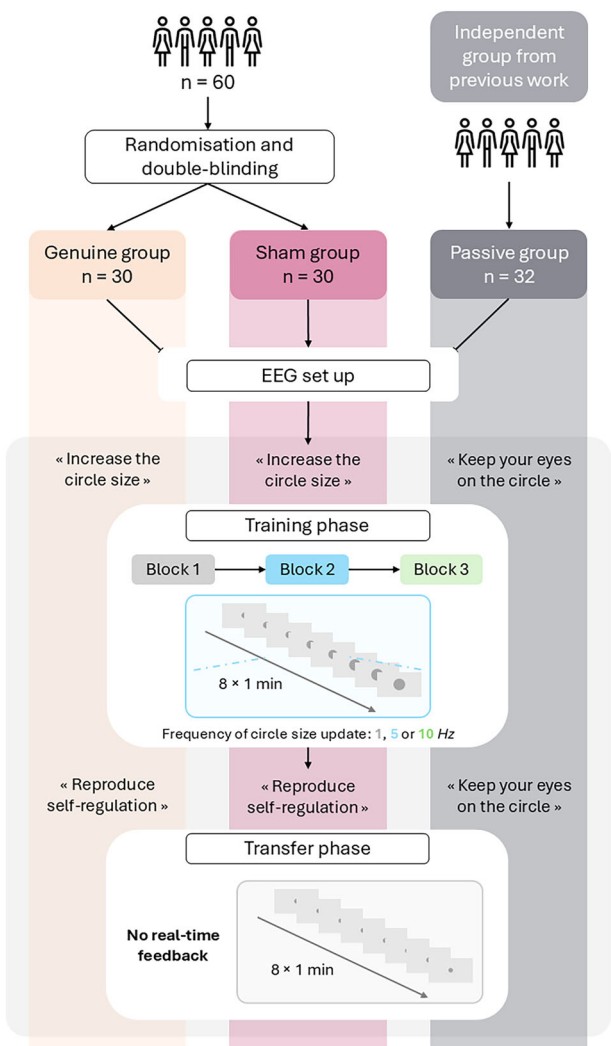

**Fig. 1 | Experimental protocol overview.** This study included three groups: a double-blind, sham-controlled EEG neurofeedback (EEG-NF) protocol with 60 healthy young adults randomly assigned to either a Genuine (*n* = 30) or Sham (*n* = 30) group, and an independent Passive group (*n* = 32) drawn from a previous study[32]. Apart from feedback veracity and verbal instructions, all three groups underwent a very similar procedure. After EEG installation, participants completed a Training phase composed of three blocks. Each block comprised eight one-minute trials. During a trial, the size of a centrally displayed grey circle was continuously updated, as neurofeedback, at either 1, 5 or 10 Hz across blocks (within-subject manipulation of feedback update frequency). In the Genuine group, participants received real-time visual feedback of their alpha (8–12 Hz) power recorded at Pz. In the Sham group, the feedback consisted of prerecorded alpha trajectories from an independent dataset[31]. Both groups were instructed to mentally increase the size of a centrally displayed grey circle. Following training, participants completed a Transfer phase, consisting of another block of eight one-minute trials. Both groups were asked to reproduce the mental strategies they identified as increasing the circle size during training (i.e., reproducing self-regulation). The Passive group followed a similar Sham protocol, but participants were not engaged in EEG-NF. Instead, they were only told to keep their eyes on the circle throughout the experimental session. This three-arm design enables isolation of: (i) the effect of genuine EEG self-regulation from non-specific influences (Genuine *vs.* Sham), (ii) the role of active engagement in self-regulation over general time-on-task effects (Sham *vs.* Passive), and (iii) the effect of timing of feedback information (1, 5 or 10 Hz) on EEG-NF learning.

## Methods

This study was preregistered before data collection on February 24th 2025, via the Open Science Framework (OSF), including research questions, hypotheses, method and analysis plan: https://osf.io/wevtz. Accordingly,

this "Methods" section closely follows the preregistration and also shares similarities with the prior work of our group[32]. To address reporting standards on the design of EEG-NF protocols, the Consensus on the reporting and experimental design of clinical and cognitive-behavioural neurofeedback studies (CRED-nf) checklist has been completed and reported in Supplementary Table 1.

## Participants and study design

A Bayesian a priori power analysis was performed to determine the required sample size for each group in the present study (see Nalborczyk et al.[35] for a similar approach). We generated simulated datasets that mirrored the planned experimental structure (i.e., matching the statistical model described by Eq. 1) while varying the sample size in each group of subjects (24, 30, 36, 42 or 48). Because confirming the null hypothesis (i.e., absence of effect) typically requires larger samples than detecting small, medium or large effects[36,37], simulations were conducted under the conservative assumption that all the considered fixed effects were null. For each candidate sample size, we estimated the average probability of obtaining a Bayes Factor of 10 or greater in support of the null hypothesis. The results indicated that even the smallest candidate sample size ($n = 24$ per group) yielded an average probability of at least 0.80 of reaching this evidential threshold.

As such, sixty healthy young adults were included using a double-blind, randomised, sham-controlled design with authorised deception. Participants were randomly assigned to a Genuine EEG-NF group ($n = 30$; $M_{age} = 22.77$ years, $SD = 3.42$, age range = 19–30; 22 females [self-reported]; 8 males [self-reported]; 24 right-handed [self-reported]) or a Sham group ($n = 30$; $M_{age} = 22.63$ years, $SD = 3.06$, age range = 19–30; 28 females [self-reported]; 2 males [self-reported]; 27 right-handed [self-reported]). Additionally, a third Passive group ($n = 32$; $M_{age} = 23.67$ years, $SD = 3.41$, age range = 18–33; 23 females [self-reported]; 9 males [self-reported]; 29 right-handed [self-reported]) was considered from a previous independent study which submitted participants to a passive visualisation task very similar to the present sham protocol (for further details, see "Independent passive group" section and the original paper[32]). This resulted in a sample size of 92 participants.

All participants reported normal or corrected-to-normal vision and no history of neurological and/or psychiatric disorders. If not reported, other demographic data (e.g., socioeconomic status, communities of descent or race/ethnicity) were not collected. Enrolment was conducted via a university learning platform and laboratory communication channels. Student participants ($n = 44$) received course credits as compensation. No individual had taken part in similar previous works. Written informed consent was obtained in accordance with the 2013 Declaration of Helsinki[38]. Each participant was anonymised using a unique code. The study received ethical approval from the French Personal Protection Committee (CPP Sud Méditerranée V, ref. 19.09.12.44636).

## Randomisation and double-blinding

Before data collection, random allocation to genuine EEG-NF or sham groups was performed using an in-house MATLAB script. Both participants and experimenters remained blind to group assignments. This double-blinding was supported by an authorised deception protocol to minimise negative psychosocial influences (e.g., motivation, expectations)[39]. Participants were informed that some elements were kept hidden for the scientific purpose of the study. Namely, they were not informed about the existence of the sham group. Group allocation remained hidden during statistical analyses.

## Material and neurofeedback implementation

All participants underwent an EEG-NF session comprising four blocks of eight 60-s trials. During each trial, a grey circle was presented at the centre of a screen. The first three blocks composed the EEG-NF training phase. The last fourth block served as a transfer phase evaluating participants' ability to self-regulate their EEG activity without real-time feedback[18]. During training, the circle size was continuously updated. Participants had to maximise

the circle size as much as possible. For the genuine EEG-NF group, the circle size was updated as a real-time feedback of the participant's alpha band (8–12 Hz) spectral power at Pz[25]. By contrast, for the sham group, the circle size was determined before data collection using alpha power fluctuations of participants from an independent study[31]. Between the training blocks of both groups, the frequency of feedback update was manipulated at 1, 5 and 10 Hz, reflecting common update frequencies in EEG-NF protocols. To counterbalance the block order across subjects, a Latin-square design was employed. All six possible triplets were generated while ensuring each frequency appears in every temporal position (123, 132, 213, 231, 312, 321). Each participant was pseudo-randomly assigned to one triplet, resulting in an even distribution within each group. Finally, during the fourth transfer block, the circle size remained constant (100 pixels). The objective was to assess participants' ability to self-regulate their EEG activity without the presentation of the feedback.

At the end of the session, participants responded to two questions: (i) "During the first three training blocks, to what extent did you feel in control of the variations of the circle?"; and (ii) "It is possible that the feedback which has been presented to you during the training was actually random. To what extent do you believe that the feedback was random?". Participants answered through five-point Likert scales (1 – "Not at all" – to 5 – "Completely"). These questions aimed to qualitatively assess participants' feelings of feedback control and belief of having received sham feedback, respectively. Importantly, these measures are recommended for implementation in sham-controlled EEG-NF studies to control for differences in feedback and blinding credibility between groups[14].

## Apparatus

The implementation of the EEG-NF session, as well as EEG data acquisition, online and offline processing were conducted in MATLAB Release 2023a (Mathworks, Inc.). Specifically, EEG data was acquired with the Brainflow library version 5-8-1[40], using an OpenBCI Cyton 8-channel board with OpenBCI Gold cup and Earclip electrodes. The genuine and sham feedback were implemented using Psychtoolbox-3[41]. They were displayed on a flat-screen computer monitor (screen resolution: 1920 × 1080 pixels; screen size: 52.704 × 29.646 cm; refresh rate: 60 Hz). The distance between the monitor and the back of the chair was kept constant (90–100 cm). Statistical analyses and graphical representations of data were performed in R version 4.3.3[42]. All graphical elements in Fig. 1 were created by the authors using Microsoft PowerPoint and MATLAB Release 2023 (for the circle display screenshots).

## EEG recording

EEG was digitalised at 250 Hz in microvolts (μV) from the OpenBCI board in MATLAB R2023a matrices. Data acquisition was done through the laptop's GPU to minimise computation time. EEG signals were recorded from six OpenBCI Gold Cup electrodes placed in accordance with the 10–20 International System at the following positions: Fp1, Fpz, Fp2, Fz, Cz, and Pz. Two OpenBCI earclip electrodes placed on the left and the right earlobes were used, respectively, as a reference for all electrodes and as a noise-cancelling ground electrode. Impedance was kept below 10 kΩ.

## Procedure

Participants were seated in front of a monitor throughout. After obtaining written and informed consent, the EEG setup was installed and impedance checked. While keeping both experimenters and participants blind, an in-house MATLAB script automatically initiated a genuine or a sham EEG-NF session. The following verbal instructions (in French) were given before the beginning of the session: "You will complete three blocks of neurofeedback training. Each block is composed of eight one-minute trials. During the eight trials of a block, a circle will be presented at the centre of the screen while continuously growing and decreasing in size. Depending on the block, its size will grow and decrease at different rates. At any given moment, the size of the circle reflects the state of your brainwaves in real-time, which are influenced by your thoughts. By keeping your eyes on the circle, your task is thus to try your best to increase as much as possible the size of the circle,

thanks to your thoughts. The goal is to find the best mental strategy that effectively renders the circle as big as possible. Over the course of the trials, you will progressively become able to self-modulate your brainwaves by intentionally modifying your thoughts and adopting the best mental strategy."

Before the transfer block, participants were also provided with verbal instructions: "Congrats for completing this training! You will now complete a final block, but without feedback. For another eight one-minute trials, a circle will be presented at the centre of the screen. However, this time, the size of the circle will not be modified. By keeping your eyes on the circle, your task will still be to try your best to self-regulate your brainwaves thanks to your thoughts. The objective is to reproduce the mental strategies you used during the training blocks to increase the circle size. Keep it up, it's almost done!" After completing this block, questions were presented to the participant for qualitative purposes (cf. "Material and neurofeedback implementation" section).

During the session, participants were also instructed to remain as calm and relaxed as possible to avoid disturbing the EEG signal. They had to press the "Enter" key to start a new trial. Experimenters remained in the same room, but out of the participant's field of vision.

## EEG online processing

During the training blocks of the genuine EEG-NF group, a time-frequency analysis (moving-window short-FFT) was executed in real-time to extract the alpha frequency band (8–12 Hz) spectral power at Pz. This procedure followed Keil et al.[43]'s recommendations (see Supplementary Table 2 for the corresponding completed checklist). At each step of the analysis, zero-phase filtering was applied using the MATLAB *filtfilt* function with a 1–20 Hz bandpass filter (4th order IIR Butterworth). A symmetric Hann window of 500 samples (2-s length) was used to compute the spectral power estimates in decibels (dB) with the MATLAB *pspectrum* function. Alpha spectral power was obtained by averaging these estimates within the 8–12 range.

Depending on the training block, different frequencies for feedback update were used: 1, 5 or 10 Hz. To match the corresponding frequency, the overlap between two consecutive windows was determined by subtracting the window sample size (500 samples) from the division of the EEG frequency sample (250 Hz) by the frequency of feedback update. This respectively resulted in window overlaps of 250 (500 – 250/1), 450 (500 – 250/5) and 475 (500 – 250/10) samples (i.e., 50%, 90% and 95%, respectively).

## EEG offline processing

EEG data was offline-processed according to Keil et al.[43]'s guidelines (see Supplementary Table 3 for the completed checklist). In-house MATLAB scripts and EEGLAB[44] were used to compute the spectral power of theta (4–8 Hz), alpha (8–12 Hz), SMR (12–15 Hz), and beta (15–30 Hz) frequency bands. Since EEG-NF studies typically lack an explicit model for the generation of oscillatory activity and the $1/f$ noise[6], we adopted the narrowband model implicitly assumed in the field[43]. Initially, data from all channels was zero-phase filtered with a 0.5 Hz high-pass filter (6th order IIR Butterworth) and a 50 Hz notch filter (2nd order IIR). The filtered data was imported into EEGLAB, where an extended Infomax Independent Component Analysis (ICA) was applied for each participant[45]. Components corresponding to eye blinks and lateral eye movements were identified and removed from the data through visual inspection of the component scalp topography, time series, and power spectra. Supplementary Table 4 presents the number of components removed for each participant. The artefact-corrected data were exported back in MATLAB format. Only data from Fz, Cz and Pz electrodes were considered for further analysis.

EEG signals were analysed in the frequency domain using the MATLAB *pspectrum* function. This function computes power spectra via FFT and employs Welch's method to enhance spectral estimates' reliability. By default, it segments the signal into as long sections as possible while ensuring a number of segments as close to (but not exceeding) 8, an overlap of 50%, and applying a Hamming window before computing the FFT. The averaged power spectra provided the final spectral estimates. Power values

for each participant and trial were then transformed to dB to promote normality in data distribution. Finally, the spectral power of theta (4–8 Hz), alpha (8–12 Hz), SMR (12–15 Hz) and beta (15–30 Hz) frequency bands was obtained by averaging the power estimates within the corresponding range.

## Independent passive group

In the previous experiment constituting the passive group[32], participants ($n = 32$) underwent a similar procedure to the sham group of the present project. The main difference lies in the instructions given to participants. Previous participants were not engaged in an EEG-NF session. The task comprised four blocks of eight one-minute trials each and was not divided into training *vs.* transfer blocks. During every trial, participants were presented with the same circle from the present project. They were steadily instructed to keep their eyes on the circle. Each block was considered as a condition and three factors were manipulated between conditions: (i) trial repetition, (ii) the continuous modification of the circle size (absent during one control condition, and present during the remaining three experimental conditions), and (iii) the frequency at which the circle was modified (either 1, 5 or 10 Hz depending on the experimental condition). Full description of the rationale, the methodology and the results can be found in the original paper[32].

To evaluate the influence of the engagement in self-regulation on targeted alpha upregulation, we considered this previous sample as a "passive" group (i.e., passive visualisation of a sham feedback) and systematically compared them to the present sham group. The same processing pipeline was applied to all groups. The same dependent variables and the same dataset dimensions were considered to enable systematic comparisons. Specifically, when comparing the current training blocks with those during which the circle was modified in previous work[32], the dataset contained the following dimensions: Subject number, Trial number, Frequency of feedback update, and Task (either genuine EEG-NF, sham EEG-NF, or passive visualisation task). When comparing the data from the transfer block with the control condition of the independent study[32], the same dimensions were considered, except for Frequency of feedback update.

## Statistical analyses and hypotheses testing

**EEG data.** The resulting twelve dependent variables (i.e., spectral power of theta, alpha, SMR and beta frequency bands measured at Fz, Cz and Pz electrodes) were standardised as z-scores across participants for statistical analyses. Alpha power at Fz, Cz and Pz was defined as the main outcome of interest, and these analyses were confirmatory. Analyses of remaining variables were exploratory. However, they served to assess the neurophysiological specificity of the current EEG-NF session, i.e., its ability to modulate alpha power. For the purpose of statistical analyses, we arbitrarily hypothesised the presence of effects of interest on all variables. First, when considering the training blocks, it implies that their spectral power is influenced by: (i) trial repetition, (ii) the frequency of feedback update, (iii) the engagement in self-regulation (i.e., sham session *vs.* passive visualisation task[32]), (iv) the veracity of the feedback (i.e., genuine *vs.* sham), and (v) each corresponding interaction effects. Second, when considering only the transfer block, it means that their spectral power is influenced by: (i) trial repetition, (ii) the engagement in self-regulation, (iii) the veracity of the feedback, and (iv) each corresponding interaction effect.

Statistical analyses were conducted using Bayesian linear multilevel models implemented with the brms[46] and rstan[47] R packages. Bayesian analyses offer multiple advantages over frequentist approaches (for extended tutorials, see Schad et al.[48,49]), including robustness in low-power contexts[37], the ability to easily construct and fit multilevel models[50], the capability to distinguish between sensitive and insensitive evidence for an absence of effect[51], and the resistance to issues arising from multiple comparison[52]. Each model treated one of the above-mentioned dependent variables as continuous, and incorporated the maximal varying effect structure to account for individual variability of participants[53]. When

analysing training blocks, each model included as fixed effects: the Trial number continuous within-subject predictor (i.e., integers from 1 to 8), the Frequency categorical within-subject predictor (i.e., 1, 5, and 10 Hz), and the Task categorical between-subject predictor (i.e., genuine EEG-NF group, sham EEG-NF group, and passive visualisation group), as well as their interactions. For the trial predictor, we defined the first trial as a reference for subsequent comparisons. For Frequency and Task predictors, we used repeated contrast matrices, which are respectively presented in Supplementary Tables 5 and 6. Specifically, assigning a repeated contrast matrix to the task predictor enabled the evaluation of both effects of the veracity of feedback (Genuine *vs.* Sham) and of the engagement in self-regulation (Sham *vs.* Passive). Regularising priors of $N(0, 1)$ were placed on each model parameter to constrain the models to plausible values and avoid overfitting issues[48]. Here is the full model equation (brms syntax):

$$Power \sim Trial * Frequency * Task + \left(Trial * Frequency \mid Subject\right) \quad (1)$$

When analysing the transfer block, we used the same procedure but only included the above-mentioned Trial and Task predictors, as well as their interaction, as fixed effects. Here is the corresponding model equation:

$$Power \sim Trial * Task + \left(Trial \mid Subject\right) \quad (2)$$

For all effects, we computed Bayes Factors (BFs) quantifying the strength of evidence for a hypothesis over another[51]. To ensure BFs' stability, we performed all reported analyses twice[49]. For each effect considered, we report the mean of the two obtained posterior distributions, as well as the largest limits of the 95% CrI. We also report the mean of the $BF_{10}$, which quantifies evidence for the alternative hypothesis over the null (i.e., presence of an effect over its absence). Following Jeffrey[54], we consider that a $BF_{10}$ of above 3 indicates substantial evidence for the alternative over the null hypothesis, and that a $BF_{10}$ of below ⅓ quantifies substantial evidence for the null over the alternative hypothesis. A $BF_{10}$ between ⅓ and 3 indicates data insensitivity to distinguish null and alternative hypotheses. When an effect was confirmed (i.e., $BF_{10} > 3$), we also reported the $BF_{10+}$, quantifying the amount of evidence for a positive-directional (i.e., one-sided) effect. Supplementary Tables 7–10 present the estimates (standardised units), 95% CrI and BFs from each of the models computed.

**Feeling of feedback control and feedback credibility.** To assess group differences in subjective experience of control and belief in random feedback variations, we conducted two Bayesian independent t-tests. The first tested ratings of perceived control over the feedback variations, and the second assessed the belief in the randomness of feedback update (cf. "Material and neurofeedback implementation" section). Each of the two variables was considered as continuous (integers from 1 to 5 from a 5-point Likert scale ranging from 1 – "Not at all" – to 5 – "Completely"). For both tests, we report the estimate and the limits of the 95% CrI. We also report the $BF_{01}$ quantifying the amount of evidence for the absence, over the presence, of an effect of the Group predictor (Genuine *vs.* Sham) on each considered dependent variable.

## Results
In this study, participants underwent an alpha EEG-NF session comprising three training blocks and one transfer block (Fig. 1). During the training blocks, participants were presented with a circle whose size either reflected their alpha (8–12 Hz) power at Pz ($n = 30$, Genuine group) or was disconnected from their EEG activity ($n = 30$, Sham group). Both groups were instructed to increase the circle size as much as possible and were not informed about the existence of the sham feedback. Additionally, this study comprises a third independent group ($n = 32$, Passive group[32]). This group underwent a similar procedure to the Sham group, but was not engaged in EEG self-regulation. Participants were solely instructed to observe visual stimulus variations.

### Alpha is upregulated regardless of the training condition
To evaluate whether the alpha power was progressively upregulated during EEG-NF training, we assessed the effect of trial repetition within a training block using Bayesian multilevel modelling[48,49]. We also evaluated the effects and corresponding interactions of trial repetition with both: (i) the veracity of feedback (i.e., comparing Genuine and Sham groups), and (ii) the engagement in self-regulation (i.e., comparing the Sham group to the independent, Passive visualisation group[32]). Both effects were included in the statistical models as one three-level 'Task' predictor (see the "Methods" section for more details).

Figure 2A presents the alpha power evolution over the training block with a 1 Hz feedback update frequency (reference level of Frequency predictor), depending on the Task to which participants were submitted (i.e., Genuine EEG-NF, Sham EEG-NF, or Passive feedback visualisation). Figure 2B displays the $BF_{10}$ obtained for the effect, on alpha power, of trial repetition and corresponding interactions with feedback veracity and engagement in self-regulation. Concerning the effect of trial repetition, extreme evidence was found in favour of a positive effect on alpha power (Fz: $\beta = 0.019$, 95% CrI [0.014, 0.025], $BF_{10} = 769175401465056$, $BF_{10+} = Inf.$; Cz: $\beta = 0.02$, 95% CrI [0.014, 0.026], $BF_{10} = 417951444524249$, $BF_{10+} = Inf.$; Pz: $\beta = 0.02$, 95% CrI [0.013, 0.026], $BF_{10} = 99443180922583$, $BF_{10+} = Inf.$). Insensitive evidence was found concerning the effect of the veracity of feedback (i.e., Genuine *vs.* Sham; Fz: $\beta = 0.331$, 95% CrI [−0.148, 0.809], $BF_{10} = 0.612$; Cz: $\beta = 0.441$, 95% CrI [−0.029, 0.909], $BF_{10} = 1.36$; Pz: $\beta = 0.512$, 95% CrI [0.056, 0.968], $BF_{10} = 2.662$), even though higher power in the Sham group is noticeable via visual inspection (see Supplementary Fig. 1). Substantial evidence supported no effect of the engagement in self-regulation (i.e., Sham *vs.* Passive; Fz: $\beta = −0.122$, 95% CrI [−0.593, 0.347], $BF_{10} = 0.272$; Cz: $\beta = −0.182$, 95% CrI [−0.642, 0.274], $BF_{10} = 0.314$; Pz: $\beta = −0.211$, 95% CrI [−0.655, 0.236], $BF_{10} = 0.236$). Importantly, extreme evidence supported the absence of interaction of trial repetition with both the veracity of feedback (Fz: $\beta = −0.001$, 95% CrI [−0.016, 0.013], $BF_{10} = 0.007$; Cz: $\beta = 0.004$, 95% CrI [−0.011, 0.019], $BF_{10} = 0.009$; Pz: $\beta = 0.002$, 95% CrI [−0.014, 0.017], $BF_{10} = 0.008$) and the engagement in self-regulation (Fz: $\beta = 0.003$, 95% CrI [−0.011, 0.017], $BF_{10} = 0.008$; Cz: $\beta = 0.003$, 95% CrI [−0.012, 0.017], $BF_{10} = 0.008$; Pz: $\beta = 0.005$, 95% CrI [−0.01, 0.02], $BF_{10} = 0.009$). Distributions of individual slopes of alpha evolution across trials are presented in Fig. 2C, depending on participant groups. Visually, all three groups exhibit similar distributions at all electrodes, with only a slightly more heterogeneous distribution in the Passive group.

Moreover, the frequency of feedback update was manipulated across training blocks (i.e., 1, 5 or 10 Hz) in all three groups. The aim was to evaluate whether this update frequency had an influence on alpha upregulation learning. The within-subject Frequency predictor and its corresponding interactions with Trial and Task were included in the statistical models as fixed effects (see Eq. 1 in the "Methods" section).

Figure 3A shows the evolution of alpha power within training depending on the frequency of feedback update. For all effects of the frequency of feedback update and corresponding interactions with Trial and Task predictors, substantial to extreme evidence supported the absence of influence on alpha power at Fz, Cz and Pz. Supplementary Table 1 provides the estimates (z-score standardised values), 95% CrI and corresponding Bayes Factors (BFs) obtained for each of the effects considered on training blocks. $BF_{10}$ quantifying evidence for effects of feedback update frequency and corresponding interactions with trial repetition are presented in Fig. 3B. Figure 3C displays very similar distributions of individual slopes of alpha power evolution across trials depending on the feedback update frequency.

Together, all these results indicate that alpha power tends to increase at Fz, Cz and Pz during the alpha EEG-NF session. Importantly, this increase is not modulated by the frequency of feedback update, the veracity of the feedback, nor the engagement of participants in self-regulation.

**Fig. 2 | Alpha power evolution within the 1 Hz training block. A** Evolution of alpha power at Fz, Cz and Pz across the trials of the 1 Hz training block (defined as the reference level of the frequency predictor). Each line point represents EEG power averaged at the group-level depending on the Task participants were submitted to (cream: Genuine EEG-NF; dark blue: Sham EEG-NF; pink red: Passive visualisation). Error bars indicate 95% confidence intervals ($n = 30$ participants for the Genuine and Sham groups; $n = 32$ participants for the Passive group). **B** Bayes Factors ($BF_{10}$) quantifying evidence for the alternative hypothesis over the null for the trial repetition effect (green) and corresponding interactions (blue: with the veracity of feedback; purple: with the engagement in self-regulation) on alpha (8–12 Hz) spectral power at Fz, Cz and Pz during the 1 Hz training block. On each plot, values are $log_{10}$-scaled, such that the solid line at $y = 1$ marks no change in evidence, with bars extending upward or downward depending on whether evidence supports the alternative or the null. Dashed lines at $y = 3$ and $y = 1/3$ indicate substantial evidence in favour of the presence or the absence of an effect, respectively. **C** Distributions of individual slopes of alpha evolution throughout the 1 Hz training block. Each point represents the trial repetition slope of one participant, depending on the undergoing Task (cream: Genuine EEG-NF; dark blue: Sham EEG-NF; pink red: Passive visualisation) and corresponding electrode (left: Fz; middle: Cz; right: Pz). Each boxplot represents the median, the first and third quartiles, along with the 95% confidence interval of the resulting distribution ($n = 30$ participants for the Genuine and Sham groups; $n = 32$ participants for the Passive group).

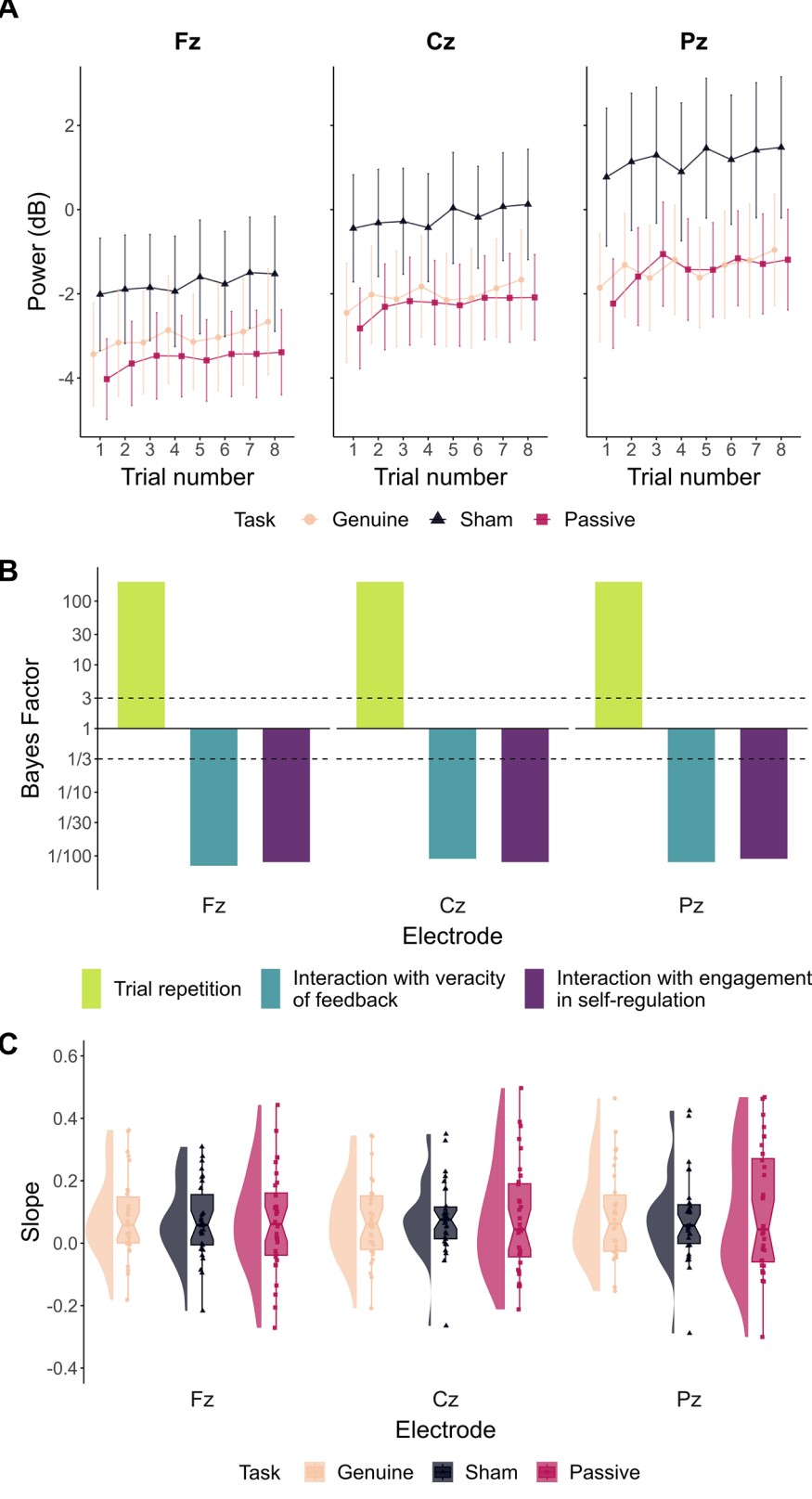

## Alpha keeps increasing without feedback

To assess the ability of participants to upregulate their alpha power without the feedback help[7,18], a transfer block without continuous feedback was implemented after the training blocks. The effects of trial repetition, the veracity of feedback, the engagement in self-regulation and corresponding interactions were evaluated through Bayesian hierarchical modelling (see Eq. 2 in the "Methods" section for more details).

Figure 4A presents the evolution of alpha power across the trials of the transfer block depending on participants' group (i.e., Genuine, Sham or Passive). $BF_{10}$ quantifying evidence for trial repetition and corresponding interaction effects are presented in Fig. 4B. We found extreme

**Article**

**Fig. 3 | Evolution of alpha power during training as a function of the frequency of feedback update. A** Evolution of alpha power at Fz, Cz and Pz across one training block of the Genuine group. Each line point represents EEG power averaged at the group-level depending on the frequency of feedback update (cream: 1 Hz; pink red: 5 Hz; dark blue: 10 Hz). Error bars indicate 95% confidence intervals ($n = 30$ participants for the Genuine and Sham groups; $n = 32$ participants for the Passive group). **B** Bayes Factors ($BF_{10}$) quantifying evidence for the alternative hypothesis over the null for the effects of feedback frequency (green: 5 Hz *vs.* 1 Hz; blue: 10 Hz *vs.* 5 Hz) and corresponding interactions with trial repetition (emerald and purple, respectively) on alpha (8–12 Hz) spectral power at Fz, Cz and Pz of the Genuine group (defined as reference level of Task predictor). On each plot, values are $log_{10}$-scaled, such that the solid line at $y = 1$ marks no change in evidence, with bars extending upward or downward depending on whether evidence supports the alternative or the null. Dashed lines at $y = 3$ and $y = 1/3$ indicate substantial evidence in favour of the presence or the absence of an effect, respectively. **C** Distributions of individual slopes of alpha evolution throughout training in the Genuine group. Each point represents the trial repetition slope of one participant, depending on the frequency of feedback update (cream: 1 Hz; pink red: 5 Hz; dark blue: 10 Hz) and corresponding electrode (left: Fz; middle: Cz; right: Pz). Each boxplot represents the median, the first and third quartiles, along with the 95% confidence interval of the resulting distribution ($n = 30$ participants for the Genuine and Sham groups; $n = 32$ participants for the Passive group).

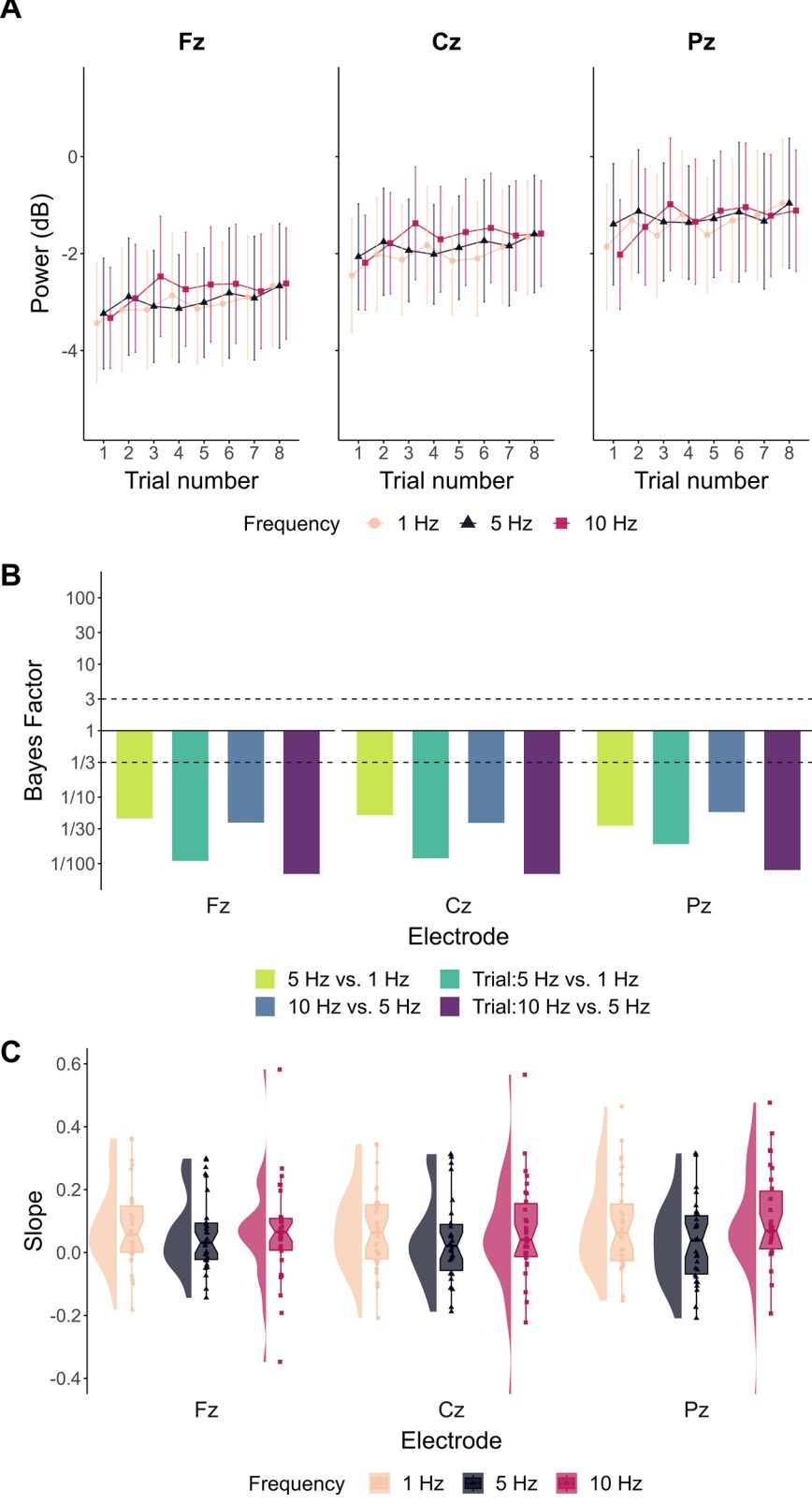

evidence supporting a positive effect of trial repetition on alpha power during the training block (Fz: $\beta = 0.033$, 95% CrI [0.022, 0.043], $BF_{10} = 1322458781605221$, $BF_{10+} = Inf.$; Cz: $\beta = 0.034$, 95% CrI [0.024, 0.045], $BF_{10} = 645504366446923$, $BF_{10+} = Inf.$; Pz: $\beta = 0.03$, 95% CrI [0.02, 0.041], $BF_{10} = 61337222253310$, $BF_{10+} = Inf.$). Insensitive evidence was obtained for the effects of the veracity of feedback (i.e., Genuine *vs.* Sham) and of the engagement in self-regulation (i.e., Sham

*vs.* Passive), despite noticeably higher absolute alpha power in the Sham group (Supplementary Fig. 1). Importantly, concerning both interactions with trial repetition, substantial to very strong evidence supporting the absence of effect was found. Supplementary Table 2 provides the estimates (z-score standardised units), 95% CrI and corresponding BFs computed for the present analyses on the transfer block. Figure 4C displays similar distributional attributes for individual slopes of alpha

**Fig. 4 | Alpha power evolution across the trials of the transfer block. A** Evolution of alpha power at Fz, Cz and Pz across the trials of the transfer block. Each line point represents EEG power averaged at the group-level depending on the Task participants were submitted to (cream: Genuine EEG-NF; dark blue: Sham EEG-NF; pink red: Passive visualisation). Error bars indicate 95% confidence intervals (*n* = 30 participants for the Genuine and Sham groups; *n* = 32 participants for the Passive group). **B** Bayes Factors ($BF_{10}$) quantifying evidence for the alternative hypothesis over the null for the trial repetition effect (green) and corresponding interactions (blue: with the veracity of feedback; purple: with the engagement in self-regulation) on alpha (8–12 Hz) spectral power at Fz, Cz and Pz during the transfer block. On each plot, values are $\log_{10}$-scaled, such that the solid line at *y* = 1 marks no change in evidence, with bars extending upward or downward depending on whether evidence supports the alternative or the null. Dashed lines at *y* = 3 and *y* = ⅓ indicate substantial evidence in favour of the presence or the absence of an effect, respectively. **C** Distributions of individual slopes of alpha evolution throughout the transfer block. Each point represents the trial repetition slope of one participant, depending on the preceding Task (cream: Genuine EEG-NF; dark blue: Sham EEG-NF; pink red: Passive visualisation) and corresponding electrode (left: Fz; middle: Cz; right: Pz). Each boxplot represents the median, the first and third quartiles, along with the 95% confidence interval of the resulting distribution (*n* = 30 participants for the Genuine and Sham groups; *n* = 32 participants for the Passive group).

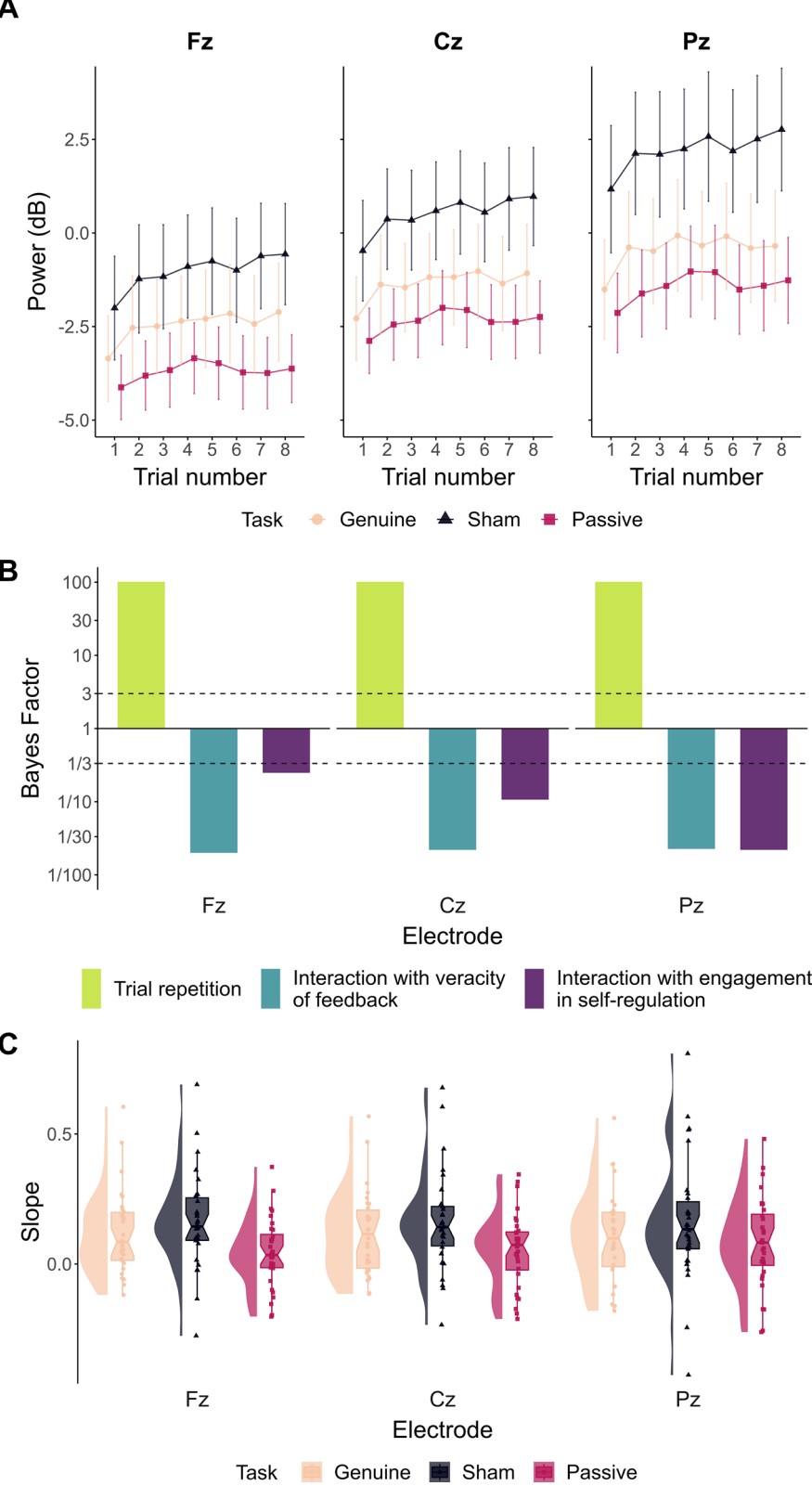

power evolution over trials for each group of participants (with, however, a slightly superior heterogeneity in the Sham group). Overall, coherently to training blocks, these results suggest an effective upregulation of alpha power without feedback, yet regardless of the veracity of the feedback during training and of the engagement in self-regulation.

## Theta and sensorimotor rhythm are also pulled upwards

We also evaluated the neurophysiological specificity of the targeted alpha upregulation by reproducing the above-described analyses on the spectral power of theta (4–8 Hz), sensorimotor rhythm (SMR; 12–15 Hz) and beta (15–30 Hz) frequency bands. Figure 5 displayed the evolution of each feature considered across the whole session.

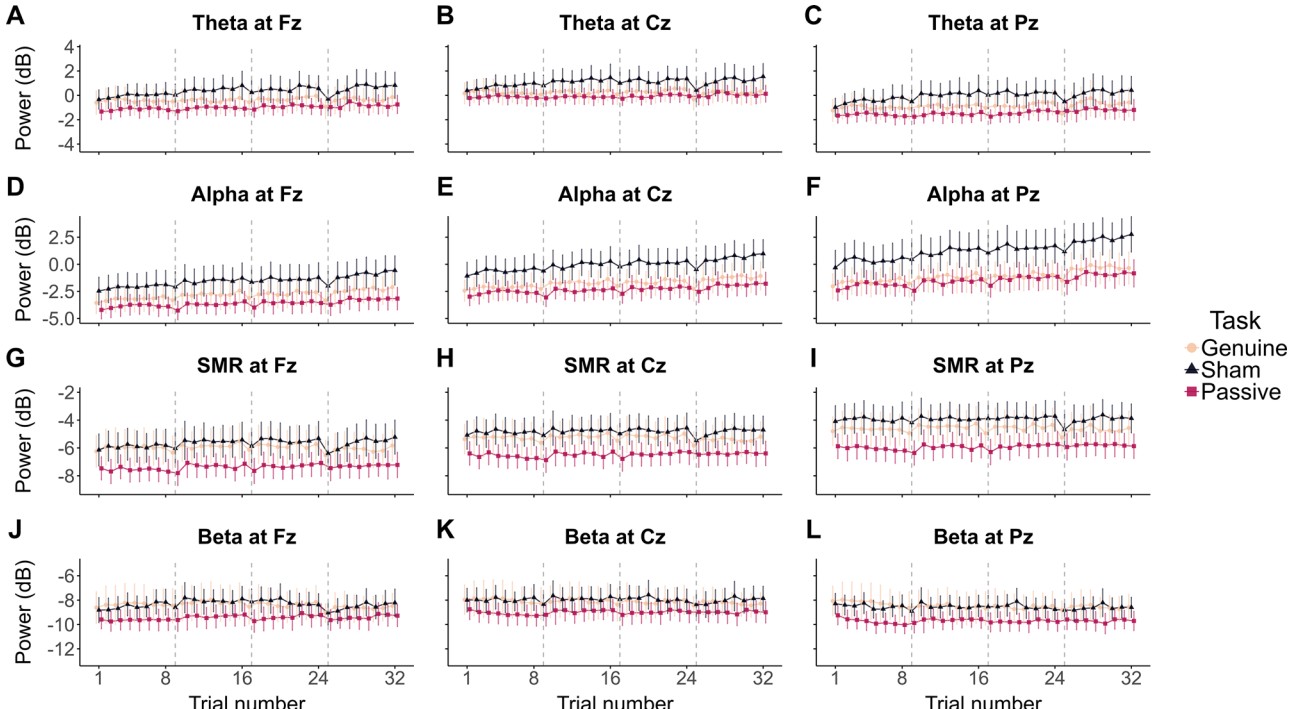

**Fig. 5 | Evolution of EEG spectral power throughout the EEG-NF session.** Evolution of theta (4–8 Hz; **A–C**), alpha (8–12 Hz; **D–F**), SMR (12–15 Hz; **G–I**) and beta (15–30 Hz; **J–L**) power across the trials of the whole EEG-NF session (no matter training *vs.* transfer phases). Each line point represents the EEG spectral power averaged at the group-level in function of the Task participants were submitted to (cream: Genuine EEG-NF session; dark blue: Sham EEG-NF session; pink red: Passive feedback visualisation). Error bars indicate 95% confidence intervals ($n = 30$ participants for the Genuine and Sham groups; $n = 32$ participants for the Passive group). Vertical dashed lines mark the first trial of each new block, the transfer block corresponding to trials 25–32.

When considering the training blocks, substantial evidence supported a positive trial repetition effect on theta power at Fz ($\beta = 0.014$, 95% CrI [0.007, 0.02], $BF_{10} = 9.146$, $BF_{10+} = 16968.7$). When considering the transfer block, a positive trial repetition effect was supported by substantial to extreme evidence for theta (Fz: $\beta = 0.029$, 95% CrI [0.018, 0.041], $BF_{10} = 49519876393386$, $BF_{10+} = Inf.$; Cz: $\beta = 0.028$, 95% CrI [0.016, 0.041], $BF_{10} = 60.8$, $BF_{10+} = 79999$; Pz: $\beta = 0.029$, 95% CrI [0.016, 0.041], $BF_{10} = 6484.6$, $BF_{10+} = Inf.$) and SMR (Fz: $\beta = 0.025$, 95% CrI [0.014, 0.036], $BF_{10} = 33.727$, $BF_{10+} = Inf.$; Cz: $\beta = 0.018$, 95% CrI [0.009, 0.027], $BF_{10} = 7.87$, $BF_{10+} = 13332.33$; Pz: $\beta = 0.018$, 95% CrI [0.009, 0.027], $BF_{10} = 9.546$, $BF_{10+} = 12635.36$) power. For most of the remaining effects during the training and the transfer blocks, substantial to extreme evidence for no effect has been obtained on all features. Few exceptions (for most we found insensitive $BF_{10}$) are presented in Supplementary Tables 3 and 4 for training and transfer blocks, respectively. Overall, these results indicate that theta (Fig. 5A–C) and SMR (Fig. 5G–I) power also exhibit, as alpha power (Fig. 5D–F) and in contrast to beta (Fig. 5J–L), a tendency to increase during the EEG-NF session.

**Genuine and sham feedback are similarly perceived**
Lastly, to assess the success of the double-blind sham-controlled procedure (i.e., participants should perceive the genuine and sham feedback similarly)[14], participants were asked at the end of the session about their feelings of control over the feedback variations and how strongly they believed that these variations were actually random. Figure 6 shows the distribution of responses depending on the veracity of the feedback (i.e., Genuine *vs.* Sham groups).

To evaluate the presence of a group difference, two Bayesian independent t-tests were run on the feeling of control over feedback variations and on the belief in random feedback variations. There was substantial evidence in favour of an absence of group effect for both (Feeling of control: $\beta = -0.028$, 95% CrI [−0.415, 0.350], $BF_{01} = 3.772$; Belief in random feedback variations: $\beta = 0.029$, 95% CrI [−0.370, −0.425], $BF_{01} = 3.776$).

## Discussion
The present study investigated the mechanisms underlying alpha upregulation over a single EEG-NF session. Alpha trajectories through training and transfer phases were evaluated as a function of the veracity of feedback (double-blind sham-controlled design), the engagement in the self-regulation (*vs.* passive feedback visualisation), and the feedback update rate (1, 5, or 10 Hz).

Findings were clear: alpha power increased consistently throughout the training and transfer phases regardless of any experimental manipulation (feedback veracity, update frequency or engagement in self-regulation). These results indicate that, contrary to prevailing assumptions in the field, EEG-NF alpha upregulation may be elicited by non-specific changes unrelated to EEG-NF settings, occurring similarly in passive sham feedback exposure.

Historically, EEG-NF studies have prioritised demonstrating the behavioural benefits of their interventions[55]. Until recently, most EEG-NF studies exclusively reported results on behavioural outcomes, implicitly assuming that improvements implied and were due to brain modulation[5]. As such, the primary EEG-NF assumption, i.e., EEG modulation occurs through a hypothesised self-regulation mechanism, has been largely overlooked[56]. A core prediction of this assumption is the occurrence of targeted EEG changes through training, as well as the superiority, in terms of EEG changes, of genuine EEG-NF over sham. Mechanistically, EEG changes are supposed to be driven by the volitional production of internal mental actions, which correlate with positive feedback (i.e., impossible to achieve in a sham protocol)[7]. Here, we therefore consider that for the targeted EEG modulation to be successful, genuine EEG-NF must produce superior EEG changes than sham. Additionally, engaging in EEG self-regulation, independently of feedback veracity, recruits numerous non-specific influences, including cognitive (e.g., high-order cognitive functions engaged through learning), psychosocial (e.g., motivation, expectations), and general repetition-related effects[14,55]. Sham protocols capture and control this wide range of non-specific factors[9]. Thus, in double-blind sham-controlled

**Fig. 6 | Behavioural responses to feedback's feeling of control and credibility.** Distribution of ratings (5-point Likert scale) of participants' feeling of control over feedback variations (**A**) and belief in the random (i.e., sham) nature of feedback variations (**B**), depending on whether the feedback was Genuine (cream) or Sham (dark blue).

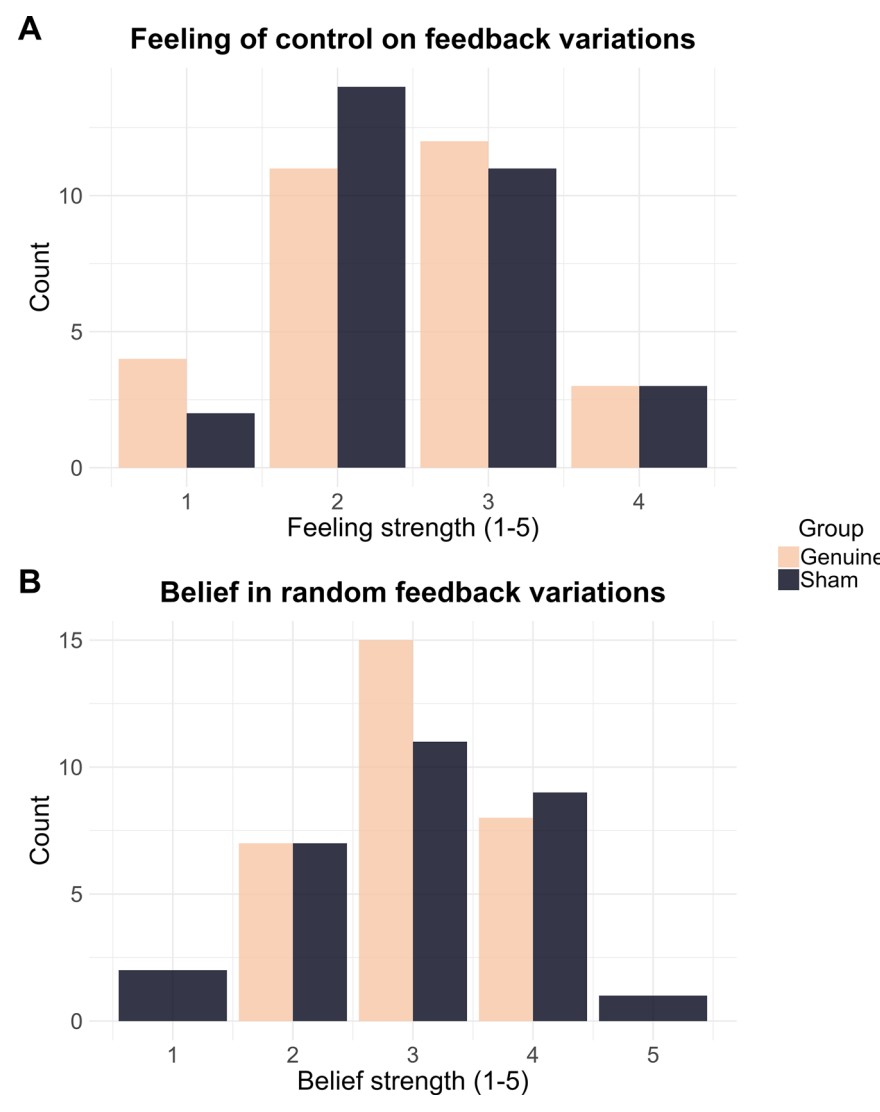

studies, if any of these factors influence EEG-NF outcomes, their non-specific nature is isolated and revealed by the sham protocol.

Yet, double-blind sham-controlled trials remain highly time-, energy- and resource-consuming[57]. As a result, their implementation is still rare in EEG-NF studies. In particular, within alpha EEG-NF research, only a few studies have properly employed such designs to isolate genuine, successful alpha upregulation from non-specific effects[26,33,57,58]. These studies did report effective alpha upregulation. However, results concerning the specificity of this modulation were conflicting. Whereas successful alpha upregulation was reported in some studies[26,33] (i.e., superiority of the genuine over the sham protocols), others demonstrated that alpha is indeed upregulated, yet identically in genuine and sham protocols[57,58]. In the present randomised double-blind design, participants completed either a genuine EEG-NF session aiming at alpha upregulation or a control, sham session. We consistently observed robust alpha upregulation during training, supported by extreme Bayesian evidence ($BF_{10} > 100$; Fig. 2). In addition, Bayesian analyses allowed us to quantify evidence for group differences in alpha evolution during training. Extreme evidence supported no difference in alpha increases in both groups during training ($BF_{10} < 1/100$; Fig. 2A, B). These Bayesian analyses align with prior null frequentist results on within-session alpha regulation[33,57,58], and, importantly, allow us to confirm the absence of any interaction between feedback veracity (genuine *vs.* sham) and trial repetition. Critically, these findings contrasted with previous claims on the possibility of easily and successfully upregulating alpha power during single EEG-NF sessions[25–27,59]. Contrary to these assumptions, they suggest that

providing genuine alpha feedback does not necessarily enable participants to achieve successful EEG self-regulation. Rather, the increases in alpha power during training appear entirely driven by non-specific factors.

Consistently, previous studies reported changes in alpha power during sham protocols or during genuine EEG-NF targeting other EEG features[22,30,60,61]. Yet, at this stage, the nature of these alpha non-stationarities remains unclear. Aside from factors implicated in EEG-NF training (notably, identified cognitive and psychosocial factors implied by self-regulation learning[55]), general fatigue and time-on-task effects are confounded across genuine and sham procedures[14]. Importantly, time-on-task effects on alpha activity are already well-documented outside of EEG-NF literature. Even in the absence of experimental manipulation, alpha power increases over time during classic EEG cognitive tasks[28,29,62,63]. These alpha increases have been associated with phenomena known to be influenced by time-on-task, such as cognitive fatigue[64] and mind wandering[62].

Accordingly, a recent study reported alpha power increases during a passive visualisation task which mimicked the temporal structure and environment of an EEG-NF session[32]. During this task, participants were presented with random visual stimulus variations. Mirroring sham feedback, a grey circle was either fixed or randomly fluctuating in size, as during the current transfer and training phases, respectively. Unlike sham protocols, participants were only instructed to observe the visual stimulus (*vs.* engaged in self-regulation, even if it is doomed to fail). Building on this, the present study evaluated whether alpha increases in this previous study would differ when submitting participants to EEG-NF. In addition to

comparing genuine and sham groups, alpha trajectories were compared between the sham group and this third, independent passive group[32]. When considering alpha evolution during EEG-NF training, extreme evidence supported that alpha upward-trajectories were the same between sham and passive groups (interaction of trial repetition and engagement in self-regulation: all $BF_{10} < 1/100$; Fig. 2). Together with the absence of difference between genuine and sham groups, these results indicate that alpha enhancement during EEG-NF may not require active engagement in self-regulation. Instead, it appears to reflect general time-on-task effects, reinforcing concerns about non-specific effects in EEG-NF studies[9,31,32].

Additionally, during training, we manipulated the frequency at which feedback was updated (1, 5 or 10 Hz) to determine whether the temporal resolution of information delivery would impact the EEG-NF learning process[18]. To our knowledge, the only prior investigation on feedback timing has manipulated the feedback delivery delay (i.e., time between power online computation and feedback delivery) rather than its update frequency[60]. This study showed that delayed feedback hampers alpha modulation efficacy. In contrast, present Bayesian analyses revealed no effect of feedback update frequency on alpha power, nor any interaction with trial repetition and group variables (i.e., veracity of feedback and engagement in self-regulation; Fig. 3). These results suggest that the frequency of feedback information does not impact alpha modulation in the present settings. However, as no specific alpha modulation occurred during training, the absence of feedback update frequency effect on alpha modulation is not surprising but reassuring.

Following training, we also implemented a transfer phase to evaluate EEG-NF learning success by assessing participants' ability to upregulate their alpha power without the feedback[14,18]. This ability ensures the acquisition of volitional self-regulation, which is thought to maximise associated behavioural effects[7]. Conversely, if participants effectively acquired self-regulation skills, successful alpha upregulation should persist without feedback. Consistent with results during training, alpha power during the transfer phase continued to exhibit an identical increase across all three groups (genuine, sham or passive; Fig. 4). These findings reinforce the conclusion that the present EEG-NF training did not allow participants to volitionally upregulate their alpha power. Although recommended[14], the implementation of transfer phases is still rare in EEG-NF studies. While a few exceptions have reported successful transfer of EEG self-regulation[65,66], those lacked adequate controls, generating biased interpretations. Here, if only the genuine EEG-NF group had been considered, we might have concluded that participants successfully upregulated their alpha power throughout training and transfer phases. Critically, the inclusion of both sham and passive groups revealed that alpha increases were independent of EEG-NF. This study highlights again the importance of implementing proper controls when evaluating EEG-NF efficacy[67].

Together, the present findings carry important implications for alpha EEG-NF protocols. Alpha EEG-NF is often considered among the most effective EEG-NF protocols[24] and associated with relaxation, meditation and cognitive performance[8,20]. Hanslmayr et al. demonstrated that EEG-NF training targeting alpha upregulation improves general cognitive performance[59], and Hardt and Kamiya found that alpha training reduces anxiety[68]. In terms of EEG modulation, alpha has been upregulated within[22,25,26,60,69,70] and between[33,60,69,70] EEG-NF sessions, as well as between resting-state measures before and after training[25,69,71]. Consequently, alpha has been considered as a key EEG-NF target that can be easily modulated. In contrast to usual multiple-session protocols, alpha upregulation has been implemented and observed within single EEG-NF sessions[25–27,71,72]. Here, we confirm effective alpha power increases during a single EEG-NF session (Fig. 5D–F). However, in this study, alpha increases appear to be spontaneous and independent of feedback veracity or participant engagement in EEG-NF[73]. As such, there may be important biases in many previous conclusions about within-session alpha EEG-NF efficacy. As previous studies typically lack proper controls[25,59,71], the EEG-NF success in modulating alpha power within a session was probably artificially driven by non-specific influences such as fatigue, mind wandering and general repetition-related

effects[32]. However, the present study (along with the independent Passive sample) did not incorporate additional specific measures of such repetition-related factors. The extent to which they effectively underlie alpha increases in EEG-NF settings, therefore, remains to be determined by further work.

One can argue that the present findings only apply to within-session EEG changes. Indeed, using a single EEG-NF session, the present study does not enable direct conclusions on other types of EEG changes (i.e., between sessions and resting-state measures). With session repetition, EEG-NF learning may be maximised and induce specific EEG modulation[6]. Accordingly, one double-blind sham-controlled study demonstrated the superiority of genuine EEG-NF over sham, in driving alpha upregulation between EEG-NF sessions[33]. However, other studies also reported alpha non-specific increases when considering between-session and resting-state changes. For instance, Naas et al. showed between-session alpha increases in both genuine alpha and sham protocols[22], and Chikhi et al. demonstrated similar increases in alpha power measured at rest before and after a genuine alpha, a genuine theta and a sham protocols[57]. Given the documented non-stationarity of alpha activity[28,29], it is very likely that alpha power also exhibits non-specific increases between resting-state measures and across EEG-NF sessions. Even if multiple sessions would have led to specific alpha enhancement in current settings, we at least insist on the necessity to implement proper controls for alpha modulation success, no matter the protocol duration and the EEG changes evaluated (within-, between-session and/or across resting-states).

Beyond the type of EEG changes used to infer successful EEG modulation, EEG-NF remains a highly heterogeneous field in which methodological choices vary widely across studies[18,74]. These include the definition of the targeted EEG feature, the definition and dynamics of the real-time feedback, the structure and duration of training trials, and the instructions or strategies provided to participants[5,8,16]. Importantly, these parameters may facilitate or hinder the emergence of the learning mechanisms that underlie successful self-regulation[75]. Such variability and the lack of protocol standardisation thus remain major obstacles for comparing outcomes across EEG-NF studies. In addition, aspects of the present EEG setup also constrain the specificity of our conclusions (limited six-channel montage). Accordingly, the present conclusions regarding the non-specificity of within-session alpha increases should be interpreted within the specific combination of protocol parameters used here. For instance, spectral densities of canonical frequency bands might reflect a complex combination of different underlying mechanisms[76,77]. As such, instead of aiming alpha power upregulation using the canonical band, some studies targeted increases in alpha power within the upper section of the alpha band (e.g., 10–12 Hz)[78,79] or while adjusting the alpha band to each participant (i.e., the so-called individual alpha frequency[33,59], although see Benwell et al.[28] acknowledging within-subject variability over time). Further work is therefore needed to draw conclusions at the field level. However, again, the present results highlight the necessity of proper control procedures in order to draw inferences about the mechanisms underlying the reported EEG changes across studies[14].

Furthermore, evaluating the occurrence of EEG changes within other EEG features is increasingly recommended to assess the neurophysiological selectivity of targeted EEG modulation[14]. We therefore reproduced all presented analyses on the spectral power of theta, SMR and beta bands. By contrast to beta power (Fig. 5J–L), we also observed increases in theta (Fig. 5A–C) and SMR (Fig. 5G–I) power during the training and transfer phases (Supplementary Tables 3 and 4). Similarly to alpha power, these increases occurred independently of feedback veracity or active engagement in self-regulation. In particular, these increases are consistent with previous work showing theta and SMR power increases in genuine and sham protocols, whether these are the EEG targets or not[13,30,33]. Interestingly, although we here confirm the stationarity of beta power in all groups, two independent studies demonstrate that beta power increases over time in genuine and sham EEG-NF protocols independently of the possibility of beta being trained[30,57]. Overall, these findings are consistent with the view that engaging in a task (even passively) may boost general EEG oscillatory activity[28]. They

stress again the relevance of taking these broadband non-stationarities into account when evaluating the evolution of EEG spectral densities over time.

More broadly, this study has important implications for the EEG-NF field. Historically, EEG-NF has focused on training oscillatory activity, particularly within defined canonical frequency bands such as theta, alpha, SMR and beta. However, oscillatory power reflects a complex mixture of neural processes that, given the present results, may not be easily accessible to the claimed volitional control[76,77]. Current evidence challenges the prevailing assumption that a single, continuous feedback signal necessarily leads to the control of unitary modifiable processes[5]. Moreover, a heterogeneous practice among EEG-NF studies is to provide participants with specific mental strategies to help them acquire volitional control over feedback[6]. Here, we did not provide them with any and encouraged them to find their own effective strategies. This approach was justified by previous reports that suggested providing strategies could counteract the learning process[18,80,81]. Importantly, no clear behaviours or mental strategies have been identified to reliably drive EEG features in a targeted direction. Without so, the premise of canonical oscillation-based EEG-NF becomes uncertain. Training participants to influence complex neurophysiological signals without an established causal mechanism may be overly optimistic. In particular, the majority of EEG-NF studies implicitly assume the narrowband model to conceive the generation of oscillatory activity, ignoring the dissociation between periodic and aperiodic components[6,43,82]. As a result, the current study, along with typical EEG-NF training protocols, presented sensory feedback to participants which reflects a mixture of both periodic and aperiodic sources. It thus remains to be determined whether the present spontaneous increases in oscillatory power would persist when controlling for changes in aperiodic activity during the training procedure and subsequent analyses (but see Kopčanová et al.[29] for spontaneous increases in periodic alpha power outside of an EEG-NF paradigm). While EEG remains a valuable and accessible tool, the current findings suggest that its use as a platform for volitional modulation of oscillatory activity warrants critical re-evaluation.

EEG-NF repeatedly claimed that its behavioural and clinical efficacy is due to targeted EEG modulation[55]. However, a robust double-blind sham-controlled study evidenced that behavioural and clinical outcomes are indeed present, but may be uniquely driven by non-specific influences[13]. Currently, some argue that EEG-NF should be used as a "superplacebo" intervention[83], while others raise ethical concerns regarding its clinical industry (use)[84,85]. Concerning brain outcomes, the field similarly claimed that targeted EEG changes are uniquely driven by an active self-regulation mechanism[6,18]. Yet, this study points out that these EEG changes can be artificially and solely induced by general repetition-related effects. Overall, contrary to, again, long-standing assumptions, classic EEG-NF protocols might not necessarily be responsible for targeted EEG changes. Cautiously, we do not claim that every EEG-NF protocol cannot drive specific EEG modulation of all features. However, this study demonstrates that claimed EEG modulation should be ensured by the systematic inclusion of proper controls.

Finally, one notable detail was that the current sham group showed marginally higher alpha power than the genuine and passive groups during training and transfer phases (Fig. 5D–F). Consistently, a previous study targeting alpha downregulation reported superior alpha amplitude in the sham compared to the genuine group, with no interaction between feedback veracity and time during training[86]. Here, the BFs evaluating for absolute group differences in power did not provide sufficient evidence for either the presence or absence of an effect (see Supplementary Tables 1 and 2). Yet, one might question whether these sham participants perceived something amiss (e.g., noticing the feedback did not respond to their efforts) and thus experienced a different state. However, participants in both sham and genuine groups reported very similar experiences over the feedback (i.e., feeling of control and belief in feedback randomness; Fig. 6). Furthermore, during the transfer block, the elevated alpha power in the sham group persisted even without any real-time feedback. If participants had somehow experienced something differently during training (which might have influenced alpha power levels), there is no apparent reason why this difference should persist without feedback. Importantly, although visual inspection suggests higher alpha power in the Sham group (Supplementary Fig. 1), Bayesian analyses confirmed all groups exhibited comparable increases in alpha power throughout the session, no matter initial power values (see Fig. 2C and Fig. 4C for individual alpha power slopes over trials). To a lesser extent, the sham group also exhibited greater power in the bands surrounding alpha throughout the session (theta: Fig. 5A–C; SMR: Fig. 5G–I). Interindividual spectral differences, especially within the alpha band, may reflect a variety of factors such as relaxation, cognitive states, and even genetics[87]. It is therefore most likely that we just got bad luck in randomising participants between groups, resulting in a higher proportion of participants with high power densities in the sham group.

In summary, this preregistered, double-blind, three-arm study demonstrates that alpha upregulation in EEG neurofeedback can occur independently of the feedback veracity and the historically claimed self-regulation mechanism[88]. The results also highlight that non-specific EEG changes may extend beyond the alpha band to broadband oscillatory activity. Rather than reflecting learned control of EEG rhythms, these spontaneous increases may stem from repetition-related factors such as fatigue and mind wandering. These results raise important considerations for EEG-NF and prompt a critical reassessment of how we interpret EEG-NF outcomes. They consequently encourage the adoption of even more rigorous controls in further research. More broadly, they also echo the call for caution when evaluating the evolution of EEG rhythms in typical EEG experiments[28,29].

## Limitations

This study has several limitations that constrain the scope of its conclusions. First, the single-session design does not allow direct inference about between-session or resting-state EEG changes. Second, although our findings highlight the contribution of spontaneous repetition-related influences such as fatigue and mind wandering, these factors were not directly measured in either the Genuine and Sham EEG-NF groups or the independent Passive group. This limits the ability to quantify their precise contribution. Third, the Passive group originates from a previous study, which was not randomised with the present Genuine and Sham groups (i.e., introducing potential differences in experimental environment between groups). Fourth, the EEG setup relied on a limited six-channel montage, which may have affected the reliability of ICA decomposition and subsequent artefact correction. Finally, EEG-NF studies encompass a huge heterogeneity in terms of protocol design, even when targeting the same EEG changes[8,16]. The present conclusions thus apply specifically to the protocol parameters employed here (i.e., definition of the alpha band, training design and structure). Further work using different protocols with similar rigorous control procedures is encouraged.

## Data availability

All data are available via the OSF at: https://osf.io/wevtz.

## Code availability

All material and analysis codes are available via the OSF at: https://osf.io/wevtz.

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

## Acknowledgements

JM was supported by a doctoral fellowship of the French Ministry of Higher Education, Research, and Innovation. VP was supported by the Fond de dotation JANSSEN HORIZON (grant numbers CNRS SPV/MB/214535). This research was supported by the Convergence Institute ILCB (ANR-16-CONV-0002), the NeuroMarseille Institute and the NeuroSchool PhD program, the Centre for Research in Education Ampiric, and the HEBBIAN ANR project (#ANR-23-CE28-0008). The funders had no role in study design, data collection and analysis, decision to publish or preparation of the manuscript.

## Author contributions

Author contributions were stated according to the Contributor Role Taxonomy (CRediT). Jacob Maaz: Conceptualisation; Data curation; Formal analysis; Investigation; Methodology; Resources; Visualisation; Writing – Original draft; Writing – Review & Editing. Laurent Waroquier: Conceptualisation; Methodology; Supervision; Writing – Review & Editing. Alexandra Dia: Data curation; Investigation. Véronique Paban: Conceptualisation; Methodology; Project administration; Resources; Supervision; Writing – Review & Editing. Arnaud Rey: Conceptualisation; Methodology; Project administration; Resources; Supervision; Writing – Review & Editing.

## Competing interests

The authors report no competing interests.
