## [Transparent Peer Review file · Communications Psychology]

Alpha power increases spontaneously during a neurofeedback session

Corresponding Author: Dr Jacob Maaz

Version 0:

Decision Letter:

Dear Mr Maaz,

Thank you for your patience during the peer-review process. Your manuscript titled "The nature of alpha modulation through neurofeedback" has now been seen by 2 reviewers, and I include their comments at the end of this message. They find your work of interest but raised some important points. We are interested in the possibility of publishing your study in Communications Psychology, but would like to consider your responses to these concerns and assess a revised manuscript before we make a final decision on publication.

We therefore invite you to revise and resubmit your manuscript, along with a point-by-point response to the reviewers. Please highlight all changes in the manuscript text file.

Editorially, we consider it important that the reviewers' concerns are fully addressed. This includes providing additional analyses where needed, improving the clarity in the methodological descriptions and figures, as well as the need to moderate conclusions when they are not directly supported by the manipulations or results.

As you revise the manuscript in response to these issues, please also implement all requests in the attached Mandatory Revision Requests document. All requirements listed in this document need to be fully met, or the work will be returned to you for further revisions without peer review. This workflow is in place to increase the likelihood that the paper will be accepted for publication. It reduces the number of rounds of revision (and review) and ensures that the reviewers vet a version of the article that is compliant with journal policies. If you have any questions regarding the required revisions, please contact the journal prior to resubmission to avoid a negative outcome.

Please submit the following items:

- Revised manuscript
- Point-by-point response to the referees' comments
- Mandatory Revision Requests Table (attached).
- Cover letter (as a separate document)

via this link: Link Redacted .

** This url links to your confidential home page and associated information about manuscripts you may have submitted or are reviewing for us. If you wish to forward this email to co-authors, please delete the link to your homepage first **

Best regards,

David Pascucci

David Pascucci
Guest Editor
Communications Psychology

REVIEWER EXPERTISE:

Reviewer #1 neural oscillations

Reviewer #2 neural oscillations

REVIEWER REPORTS:

Reviewer #1 (Remarks to the Author):

Summary

In this randomized double-blind study, the authors employed two control groups to control for unspecific effects of neurofeedback training of the alpha rhythm of the EEG. The authors observed a robust increase of the alpha, theta and SMR rhythms across blocks of a single session of training at three different locations on the scalp. The increase was comparable in all groups regardless of the nature of feedback or engagement with the task, revealing the existence of unspecific and trivial modulatory effects.

Evaluation

The study is well conceived, timely, and highly relevant for the research and clinical practice of neurofeedback. Statistical analysis of very informative as well as all the graphs and the picture. Methods for the analysis of EEG data are also compatible with the highest standards. Before I can recommend it for publication there are a few points to be better explained in the text, which I describe below:

Major points

In the methods, the authors speak about four blocks of activity, the pictures show 8 of them. It is not clear to me how many independent blocks of training were applied and how long was their duration.

The effects reported by the authors are compatible with increasing relaxation, which comes from the familiarity of the participants with the training situation. This is suggestive about the need to aggregate further levels of control to existing neurofeedback protocols in order to investigate its effectiveness.

I do not understand what is meant under the frequency of feedback update 1, 5, and 10 Hz. Did you provide feedback to a different frequency band during some intervals or is this the frequency with which the circle could change?

Why are the results for theta, beta and SMR represented in a different temporal scale than the other figures? Why do you not show all the plots using training block as a unit for the x-axis?

Minor points

To increase the informativeness of the study, the proportion of EEG recording affected by artifacts should be calculated and reported.

Reviewer #2 (Remarks to the Author):

Manuscript Number: COMMSPSYCHOL-25-0712

Review of „ The nature of alpha modulation through neurofeedback “

The manuscript reports a preregistered, double-blind, sham-controlled EEG neurofeedback (EEG-NF) study, incorporating an additional passive control group. The authors of the study investigated whether alpha upregulation during a single EEG-NF session in healthy adults reflects genuine volitional control enabled by neurofeedback, or unspecific factors such as time-on-task. In the present study, the effects of three between-subject groups (Genuine NF, Sham NF, Passive

Visualisation), three training blocks with visual feedback, and one transfer block without feedback, they observe robust increases in alpha power over trials at frontal, central and parietal electrodes, as well as parallel increases in other frequency bands e.g. in theta, SMR and beta band. Bayesian multilevel models provide strong evidence for time-on-task effects and for the absence of Group \times Trial interactions. The authors conclude that alpha upregulation is “purely non-specific”, does not depend on feedback veracity or active self-regulation, and that foundational assumptions of EEG-NF should be reconsidered.

The central question is important and highly relevant for the EEG-NF field, and the study has several strengths (preregistration, double-blind sham control, inclusion of a passive group, use of Bayesian modelling). However, several aspects of the neurofeedback implementation, EEG methodology, and the mechanistic interpretation require further clarification and a more cautious framing. Hence, in its current form I find that the strength and scope of the main claims exceed what the data and the specific protocol can support. With a more cautious framing of the main claims, a clearer restriction of their scope, and a somewhat deeper discussion of the methodological limitations, I believe the manuscript could make a strong and constructive contribution to ongoing debates about EEG-NF mechanisms and best practices.

Below are my main concerns and suggestions for improvement.

Major comments:

1. Scope and strength of the conclusions versus the actual evidence

The authors repeatedly uses very strong wording, e.g. that alpha EEG-NF upregulation is “purely non-specific”, that the study “challenges the foundational assumption of EEG-NF”, and that behavioural/clinical outcomes are “uniquely driven” by non-specific influences. In my opinion, these formulations go beyond what the data justify.

From the Bayesian analyses the authors show that (i) There is very strong evidence for a global time-on-task effect on alpha (and other bands) across trials. (ii) There is extreme evidence for the absence of Group \times Trial interactions (Genuine vs. Sham vs. Passive), i.e. learning slopes are similar across conditions. (iii) For group main effects (absolute power differences between groups), however, the authors themselves describe the evidence as “insensitive” (BFs near 1), and at least at Pz the credible interval for the “veracity” effect does not include zero.

The data demonstrate that increases in band power occur in all three groups during trials, and that the slopes of these trials do not differ between conditions. The evidence provided does not convincingly demonstrate that alpha upregulation is “purely non-specific” or that any and all group differences can be ruled out. I would strongly encourage the authors to: (i) soften field-level statements (“purely”, “uniquely”, “foundational assumption”), and frame the results more cautiously, e.g. “as evidence that under the present single-session protocol with coarse band-limited feedback at Pz, alpha increases are dominated by non-specific time-on-task effects.”.

In my view from the present study the authors have over-generalised from a highly specific setting, a fact that is also made in the discussion. Again, the specific setting is (i) a single session was conducted. (ii) coarse band-limited 8–12 Hz power at Pz was used for feedback from a 6-channel scalp montage. (iii) Very simple monotonic visual feedback was provided (“try to increase a grey circle”). I think the authors showed that there are several relevant protocols which might have shown more specific “volitional control” related effects, which are not discussed, including “multi-session”, “source-/spatial-filter-based” and “individual-alpha-peak (IAF)-based” protocols. It is my opinion that the conclusions should be explicitly limited to this protocol and target signal, and clearly distinguished from that which is currently unknown for other, more specific EEG-NF approaches.

2. Mechanistic and functional interpretation of “alpha upregulation”

The manuscript advances a series of mechanistic claims that are not always supported by empirical evidence. For instance, it suggests that “spontaneous repetition-related processes” are the primary drivers, and that “no true self-regulation” exists. However, the manuscript fails to provide quantifiable data to support these claims, and the functional meaning of alpha upregulation in this paradigm remains unclear.

The task is very low-engagement by design: Participants stare at a grey circle for multiple 1-minute trials and try to maximise its size. There is no real task or performance metric, and no behavioural outcome. This setup is almost optimally suited to elicit classic time-on-task phenomena (fatigue, vigilance decline, mind-wandering), which are well known to increase alpha and theta power over time. Did the authors explicitly assess whether participants actively engaged in a (specific) mental strategy to perform the task and reach the instructed goal? How can the authors guarantee that the participants themselves were able to gain real control in the “genuine NF” condition without a clear outcome, not only in terms of power modulation across the trial blocks or the “transfer block”?

The Discussion attributes the observed spectral drifts to “repetition-related processes”, “fatigue” and “mind-wandering”, yet none of these is actually measured (vigilance scales, mind-wandering scales, etc.). As a result, the interpretation remains speculative: the data clearly show non-stationary bandpower, but they do not distinguish between different underlying processes.

The functional meaning of “alpha upregulation” in this specific protocol remains unclear. Alpha has been associated with a number of cognitive processes, including inhibitory gating, cortical “idling”, mind-wandering and relaxation. The authors do not provide a clear definition of what is meant by the term “successful alpha upregulation” in this context. Absent behavioural or cognitive measures, the concept of “ability to upregulate alpha” remains at a purely signal-level notion.

I would suggest that the authors clearly distinguish the findings at the EEG feature level (time-dependent drift, lack of condition-specific learning) from any conclusions regarding mechanisms and functional consequences. Furthermore, it is advised that the “mechanistic language” be substantially moderated, unless additional behavioural/outcomes measures can be provided. As demonstrated in the study, there is clear evidence of time-on-task–related non-stationarity of bandpower under these specific NF-like conditions. However, the study is less clear in its ability to distinguish between “true” volitional self-regulation and “spontaneous repetition-related processes” in a mechanistic sense.

3. Neurofeedback implementation and opportunity for learning

The specific neurofeedback implementation appears to be well suited to detecting drift in EEG-scalp bandpower. However, in my view that it is not optimally designed to provide volitional self-regulation with a fair opportunity to succeed.

Continuous 1-minute trials without explicit baseline vs. regulation phases: Each training trial consists of 60 s of continuous feedback, during which participants are told to make the circle as large as possible. Unlike many NF protocols, there are no clearly defined “baseline” vs. “regulation” periods within a trial or a clear per-trial reference against which modulation is evaluated. Thus, participants have little temporal structure to infer which mental changes drove feedback changes. At the same time, slow non-specific drifts are fully reflected in the feedback signal.

Online signal processing is based on absolute bandpower at Pz: Online feedback uses 2-s Hann windows, 1–20 Hz band-pass filtering and average 8–12 Hz power at Pz in dB. There is no mention of a moving baseline or any drift correction. This means that slow changes in global state (vigilance, mind-wandering or any possible signal artefacts) are directly mapped to the circle, while subtle, faster modulations might be comparatively hard to detect.

Thus, the current protocol seems almost optimised to show that band-limited Pz power drifts over time independently of feedback veracity, while perhaps underestimating what might be possible under more supportive training conditions (clearer trial-phase structure, relative feedback, more engaging tasks). I think the authors should make this limitation explicit and be cautious in extrapolating from this specific NF design to claims about “true self-regulation” in general.

4. EEG measurement, spatial specificity and spectral modelling

Several aspects of the EEG setup and analysis limit the specificity of the conclusions.

Very limited montage and ICA: EEG was recorded from only six electrodes (Fp1, Fpz, Fp2, Fz, Cz, Pz), and extended Infomax ICA was applied to remove ocular artefacts. In view of the limited number of channels, ICA-based separation of ocular and neuronal sources is likely to be less reliable. Furthermore, there is a non-negligible risk of inadvertently removing neural activity.

Scalp bandpower at Pz as a mixed, non-specific target: Feedback is based on 8–12 Hz power at Pz, which is a linear mixture of many cortical and non-cortical sources. Even if participants had acquired some control over a subset of relevant sources, this may not be visible in this mixed signal. Conversely, any apparent control over this measure would be non-specific with respect to underlying generators. The present study therefore speaks most directly to coarse, single-electrode bandpower targets, not to more spatially specific or source-based NF protocols (or even showing some of the offline analysis in the source space). If analysis are not possible at least need to be acknowledged in the discussion.

Considering individual alpha peak (IAF): The study uses a fixed 8–12 Hz definition of alpha and does not consider individual alpha peak frequency, despite literature suggesting that IAF properties can be important for alpha/SMR modulation and volitional control in brain-computer interface performance – see for example B. Blankertz et al. (2010) Neurophysiological predictor of SMR-based BCI performance, *Neuroimage* 2010 Jul 15;51(4):1303-9. It would be helpful if the authors acknowledged this and clarified that their conclusions apply to band-defined alpha at Pz, not necessarily to IAF-centred protocols

Narrowband model vs. “broadband non-stationarities”: The Discussion rightly highlights the importance of broadband non-stationarity, yet the analyses are based on classical narrowband power without explicit separation between periodic (oscillatory peaks) and aperiodic 1/f components. Methods such as spectral parametrization (e.g. FOOOF-like approaches) would allow one to test whether the observed alpha “upregulation” persists after controlling for changes in the aperiodic background – see for example T. Donoghue et al. (2020). Parameterizing neural power spectra into periodic and aperiodic components. *Nature Neuroscience*, 23, 1655-1665. Given how central the “broadband” argument is for the interpretation, I would encourage such an analysis. Otherwise, the authors have to make an explicit statement that periodic vs. aperiodic contributions cannot be disentangled in the current data – this may have implications for the overall validity of the conclusion drawn.

5. Use and interpretation of the passive control group

The conceptual logic behind the three groups (Genuine vs. Sham vs. Passive) is attractive: contrasting specific self-regulation, engagement without veridical feedback, and mere exposure/time-on-task. However, the empirical implementation raises some issues: (i) The Passive group is drawn from a previous study, not randomised within the same experiment. This introduces potential differences in context, recruitment, and experimental environment that are not controlled for. (ii) There are no detailed subjective measures of engagement, boredom, or fatigue or any other outcome behavioral outcome measure of control over alpha power during the trials that would allow a fine-grained comparison of mental state between Sham and Passive groups. While I agree that the pattern is consistent with a strong contribution of time-on-task, I would recommend softening this language and explicitly acknowledging the limitations of using an independent passive sample and the lack of detailed subjective state measures.

Version 1:

Decision Letter:

Dear Mr Maaz,

Your manuscript titled "Alpha power increases spontaneously during a neurofeedback session" has now been seen by our reviewers, whose comments appear below. In light of their advice I am delighted to say that we are happy, in principle, to publish a suitably revised version in Communications Psychology.

We therefore invite you to revise your paper one last time to address the remaining concerns of our reviewers and a list of editorial requests. At the same time we ask that you edit your manuscript to comply with our format requirements and to maximise the accessibility and therefore the impact of your work.

EDITORIAL REQUESTS:

SUBMISSION INFORMATION:

OPEN ACCESS:

* DATA AVAILABILITY:

Link Redacted

Best regards,

David Pascucci

David Pascucci
Guest Editor
Communications Psychology

REVIEWERS' COMMENTS:

Reviewer #2 (Remarks to the Author):

Thanks for the revised manuscript and the improved discussion and the new limitations section. Overall, the revision has adequately addressed my major concerns, and I recommend the manuscript for acceptance.

** Visit Nature Research's author and referees' website at http://www.nature.com/authors

href="http://www.nature.com/authors">www.nature.com/authors for information about policies, services and author benefits**

Editor: David Pascucci

Dear Prof. Pascucci,

We sincerely thank you and the reviewers for your interest, work and insightful comments. Please find below our responses to the concerns raised by the reviewers. Note that the Reviewers' comments are in black and our responses in blue.

Please also find attached our revised manuscript and our revised supplementary material, where each new change is highlighted in yellow. Also attached is the required checklist document 'Mandatory Revision Requests', where each point which necessitated a change has been ticked in the right-hand column.

On behalf of all co-authors,
Sincerely,

The corresponding author

Reviewers' reports:

Reviewer #1

REMARKS TO THE AUTHORS

Summary

In this randomized double-blind study, the authors employed two control groups to control for unspecific effects of neurofeedback training of the alpha rhythm of the EEG. The authors observed a robust increase of the alpha, theta and SMR rhythms across blocks of a single session of training at three different locations on the scalp. The increase was comparable in all groups regardless of the nature of feedback or engagement with the task, revealing the existence of unspecific and trivial modulatory effects.

Evaluation

The study is well conceived, timely, and highly relevant for the research and clinical practice of neurofeedback. Statistical analysis of very informative as well as all the graphs and the picture. Methods for the analysis of EEG data are also compatible with the highest standards. Before I can recommend it for publication there are a few points to be better explained in the text, which I describe below:

Major points

In the methods, the authors speak about four blocks of activity, the pictures show 8 of them. It is not clear to me how many independent blocks of training were applied and how long was their duration.

Thank you for your reviews. Participants were indeed submitted to four blocks of activity (each blocks comprising 8 trials). The first 3 blocks composed the training phase and the last one the transfer phase. We strengthen this description in the legend of Fig. 1, as well as in the Method section (see new lines 143-144, 209-212 and 222 of the revised manuscript).

¹ In accordance with the Mandatory Revision Requests from the Editorial Board, the title is now modified:
"Alpha power increases spontaneously during a neurofeedback session"

The effects reported by the authors are compatible with increasing relaxation, which comes from the familiarity of the participants with the training situation. This is suggestive about the need to aggregate further levels of control to existing neurofeedback protocols in order to investigate its effectiveness.

I do not understand what is meant under the frequency of feedback update 1, 5, and 10 Hz. Did you provide feedback to a different frequency band during some intervals or is this the frequency with which the circle could change?

This frequency corresponds to the rate at which the circle changed during training. Clarified in new lines 122-124.

Why are the results for theta, beta and SMR represented in a different temporal scale than the other figures? Why do you not show all the plots using training block as a unit for the x-axis?

Since the results on theta, SMR and beta are of secondary interest in this study, the aim of Fig. 5 was to present at once the evolution of each band power. Additionally, it enables readers to appreciate alpha evolution throughout the task (not restricted to one block of training or transfer). For clarity, we specified in figure legend that the last (third) dashed line of each plot corresponds to the first trial of the transfer block (see new lines 579-580).

Minor points

To increase the informativeness of the study, the proportion of EEG recording affected by artifacts should be calculated and reported.

An ICA (through EEGLAB) has been performed on the data of each participant. Resulting ICs which corresponded to ocular artifacts were manually identified for correction through visual inspection of the component scalp topography, time series, and power spectra. Then, these ICs were rejected from the data (with the 'pop_subcomp' EEGLAB function). However, to our knowledge, such a procedure does not provide the proportion of EEG recordings rejected for each participant. Yet, the number of rejected ICs by participant is presented in new Supplementary Table 4.

Reviewer #2:

The manuscript reports a preregistered, double-blind, sham-controlled EEG neurofeedback (EEG-NF) study, incorporating an additional passive control group. The authors of the study investigated whether alpha upregulation during a single EEG-NF session in healthy adults reflects genuine volitional control enabled by neurofeedback, or unspecific factors such as time-on-task. In the present study, the effects of three between-subject groups (Genuine NF, Sham NF, Passive Visualisation), three training blocks with visual feedback, and one transfer block without feedback, they observe robust increases in alpha power over trials at frontal, central and parietal electrodes, as well as parallel increases in other frequency bands e.g. in theta, SMR and beta band. Bayesian multilevel models provide strong evidence for time-on-task effects and for the absence of Group \times Trial interactions. The authors conclude that alpha upregulation is "purely non-specific", does not depend on feedback veracity or active self-regulation, and that foundational assumptions of EEG-NF should be reconsidered.

The central question is important and highly relevant for the EEG-NF field, and the study has several strengths (preregistration, double-blind sham control, inclusion of a passive group, use of Bayesian modelling). However, several aspects of the neurofeedback implementation, EEG methodology, and the mechanistic interpretation require further clarification and a more cautious framing. Hence, in its current form I find that the strength and scope of the main claims exceed what the data and the specific protocol can support. With a more cautious framing of the main claims, a clearer restriction of their scope, and a somewhat deeper discussion of the methodological limitations, I believe the manuscript could make a strong and constructive contribution to ongoing debates about EEG-NF mechanisms and best practices.

Below are my main concerns and suggestions for improvement.

Major comments:

1. Scope and strength of the conclusions versus the actual evidence

The authors repeatedly uses very strong wording, e.g. that alpha EEG-NF upregulation is "purely non-specific", that the study "challenges the foundational assumption of EEG-NF", and that behavioural/clinical outcomes are "uniquely driven" by non-specific influences. In my opinion, these formulations go beyond what the data justify.

From the Bayesian analyses the authors show that (i) There is very strong evidence for a global time-on-task effect on alpha (and other bands) across trials. (ii) There is extreme evidence for the absence of Group \times Trial interactions (Genuine vs. Sham vs. Passive), i.e. learning slopes are similar across conditions. (iii) For group main effects (absolute power differences between groups), however, the authors themselves describe the evidence as "insensitive" (BFs near 1), and at least at Pz the credible interval for the "veracity" effect does not include zero.

The data demonstrate that increases in band power occur in all three groups during trials, and that the slopes of these trials do not differ between conditions. The evidence provided does not convincingly demonstrate that alpha upregulation is "purely non-specific" or that any and all group differences can be ruled out. I would strongly encourage the authors

to: (i) soften field-level statements (“purely”, “uniquely”, “foundational assumption”), and frame the results more cautiously, e.g. “as evidence that under the present single-session protocol with coarse band-limited feedback at Pz, alpha increases are dominated by non-specific time-on-task effects.”.

In my view from the present study the authors have over-generalised from a highly specific setting, a fact that is also made in the discussion. Again, the specific setting is (i) a single session was conducted. (ii) coarse band-limited 8–12 Hz power at Pz was used for feedback from a 6-channel scalp montage. (iii) Very simple monotonic visual feedback was provided (“try to increase a grey circle”). I think the authors showed that there are several relevant protocols which might have shown more specific “volitional control” related effects, which are not discussed, including “multi-session”, “source-/spatial-filter-based” and “individual-alpha-peak (IAF)-based” protocols. It is my opinion that the conclusions should be explicitly limited to this protocol and target signal, and clearly distinguished from that which is currently unknown for other, more specific EEG-NF approaches.

Thank you for your precise and informative reviews. We overall align with the limits in terms of mechanistic interpretations. We soften the claims acknowledged in the revised discussion (see new lines 606, 642-643, 669, 691-692, 698, 709-711, 776, 798, 804-806, and 836). Additionally, in accordance with the journal policy, we now summarise main limitations in a dedicated subsection at the end of the revised discussion (see new lines 842-856).

2. Mechanistic and functional interpretation of “alpha upregulation”

The manuscript advances a series of mechanistic claims that are not always supported by empirical evidence. For instance, it suggests that “spontaneous repetition-related processes” are the primary drivers, and that “no true self-regulation” exists. However, the manuscript fails to provide quantifiable data to support these claims, and the functional meaning of alpha upregulation in this paradigm remains unclear.

The task is very low-engagement by design: Participants stare at a grey circle for multiple 1-minute trials and try to maximise its size. There is no real task or performance metric, and no behavioural outcome. This setup is almost optimally suited to elicit classic time-on-task phenomena (fatigue, vigilance decline, mind-wandering), which are well known to increase alpha and theta power over time. Did the authors explicitly assess whether participants actively engaged in a (specific) mental strategy to perform the task and reach the instructed goal? How can the authors guarantee that the participants themselves were able to gain real control in the “genuine NF” condition without a clear outcome, not only in terms of power modulation across the trial blocks or the “transfer block”?

The Discussion attributes the observed spectral drifts to “repetition-related processes”, “fatigue” and “mind-wandering”, yet none of these is actually measured (vigilance scales, mind-wandering scales, etc.). As a result, the interpretation remains speculative: the data clearly show non-stationary bandpower, but they do not distinguish between different underlying processes.

The functional meaning of “alpha upregulation” in this specific protocol remains unclear. Alpha has been associated with a number of cognitive processes, including inhibitory gating, cortical “idling”, mind-wandering and relaxation. The authors do not provide a clear definition of what is meant by the term “successful alpha upregulation” in this

context. Absent behavioural or cognitive measures, the concept of "ability to upregulate alpha" remains at a purely signal-level notion.

I would suggest that the authors clearly distinguish the findings at the EEG feature level (time-dependent drift, lack of condition-specific learning) from any conclusions regarding mechanisms and functional consequences. Furthermore, it is advised that the "mechanistic language" be substantially moderated, unless additional behavioural/outcomes measures can be provided. As demonstrated in the study, there is clear evidence of time-on-task-related non-stationarity of bandpower under these specific NF-like conditions. However, the study is less clear in its ability to distinguish between "true" volitional self-regulation and "spontaneous repetition-related processes" in a mechanistic sense.

Thank you again for your comments. Here are our responses to the points raised:

- In terms of EEG changes, the systematic absence of interaction between Task and Trial suggests that alpha increases is driven by spontaneous repetition-related influences in the specific context of the study. We agree that this conclusion remains to be generalized to other EEG-NF protocols (in terms of EEG changes, targets and design). See new lines 720-755.
- Additionally, this interpretation indeed holds on the assumption that participants were engaged in the self-regulation task within the Genuine and Sham protocols (as there is no additional direct measures of active engagement). We now acknowledge this limitation in the revised discussion (see new lines 716-719).
- Functional meaning of alpha upregulation: now explicitly defined (see new lines 614-618, 628-631).

3. Neurofeedback implementation and opportunity for learning

The specific neurofeedback implementation appears to be well suited to detecting drift in EEG-scalp bandpower. However, in my view that it is not optimally designed to provide volitional self-regulation with a fair opportunity to succeed.

Continuous 1-minute trials without explicit baseline vs. regulation phases: Each training trial consists of 60 s of continuous feedback, during which participants are told to make the circle as large as possible. Unlike many NF protocols, there are no clearly defined "baseline" vs. "regulation" periods within a trial or a clear per-trial reference against which modulation is evaluated. Thus, participants have little temporal structure to infer which mental changes drove feedback changes. At the same time, slow non-specific drifts are fully reflected in the feedback signal.

Online signal processing is based on absolute bandpower at Pz: Online feedback uses 2-s Hann windows, 1–20 Hz band-pass filtering and average 8–12 Hz power at Pz in dB. There is no mention of a moving baseline or any drift correction. This means that slow changes in global state (vigilance, mind-wandering or any possible signal artefacts) are directly mapped to the circle, while subtle, faster modulations might be comparatively hard to detect.

Thus, the current protocol seems almost optimised to show that band-limited Pz power drifts over time independently of feedback veracity, while perhaps underestimating what might be possible under more supportive training conditions (clearer trial-phase structure, relative feedback, more engaging tasks). I think the authors should make this limitation explicit and be cautious in extrapolating from this specific NF design to claims about "true self-regulation" in general.

This limitation is now acknowledged (along with the last concerns addressed above; see new lines 720-755 and).

4. EEG measurement, spatial specificity and spectral modelling

Several aspects of the EEG setup and analysis limit the specificity of the conclusions. Very limited montage and ICA: EEG was recorded from only six electrodes (Fp1, Fpz, Fp2, Fz, Cz, Pz), and extended Infomax ICA was applied to remove ocular artefacts. In view of the limited number of channels, ICA-based separation of ocular and neuronal sources is likely to be less reliable. Furthermore, there is a non-negligible risk of inadvertently removing neural activity.

Done (see new lines 743-744 and 851-852).

Scalp bandpower at Pz as a mixed, non-specific target: Feedback is based on 8–12 Hz power at Pz, which is a linear mixture of many cortical and non-cortical sources. Even if participants had acquired some control over a subset of relevant sources, this may not be visible in this mixed signal. Conversely, any apparent control over this measure would be non-specific with respect to underlying generators. The present study therefore speaks most directly to coarse, single-electrode bandpower targets, not to more spatially specific or source-based NF protocols (or even showing some of the offline analysis in the source space). If analysis are not possible at least need to be acknowledged in the discussion. Considering individual alpha peak (IAF): The study uses a fixed 8–12 Hz definition of alpha and does not consider individual alpha peak frequency, despite literature suggesting that IAF properties can be important for alpha/SMR modulation and volitional control in brain-computer interface performance – see for example B. Blankertz et al. (2010) Neurophysiological predictor of SMR-based BCI performance, *Neuroimage* 2010 Jul 15;51 (4):1303-9. It would be helpful if the authors acknowledged this and clarified that their conclusions apply to band-defined alpha at Pz, not necessarily to IAF-centred protocols

Done (see new lines 743-755).

Narrowband model vs. “broadband non-stationarities”: The Discussion rightly highlights the importance of broadband non-stationarity, yet the analyses are based on classical narrowband power without explicit separation between periodic (oscillatory peaks) and aperiodic 1/f components. Methods such as spectral parametrization (e.g. FOOOF-like approaches) would allow one to test whether the observed alpha “upregulation” persists after controlling for changes in the aperiodic background – see for example T. Donoghue et al. (2020). Parameterizing neural power spectra into periodic and aperiodic components. *Nature Neuroscience*, 23, 1655-1665. Given how central the “broadband” argument is for the interpretation, I would encourage such an analysis. Otherwise, the authors have to make an explicit statement that periodic vs. aperiodic contributions cannot be disentangled in the current data – this may have implications for the overall validity of the conclusion drawn.

Thanks for this relevant remark. However, the design of the present training phase was based on a narrowband model (adopted by all EEG-NF studies, to the best of our knowledge). As such, the feedback sent to participant reflected both periodic and aperiodic components of canonical alpha band power. Doing this analysis offline would thus not reflect a feature participants tried to control. We now address this limitation in the revised discussion (see new lines 785-793). We also acknowledge a recent study, outside

of the EEG-NF literature, which quantified spontaneous increases in periodic alpha over time: Kopčanová, M., Thut, G., Benwell, C. S. Y., & Keitel, C. (2025). Characterising time-on-task effects on oscillatory and aperiodic EEG components and their co-variation with visual task performance. *Imaging Neuroscience*, 3, imag_a_00566.

5. Use and interpretation of the passive control group

The conceptual logic behind the three groups (Genuine vs. Sham vs. Passive) is attractive: contrasting specific self-regulation, engagement without veridical feedback, and mere exposure/time-on-task. However, the empirical implementation raises some issues: (i) The Passive group is drawn from a previous study, not randomised within the same experiment. This introduces potential differences in context, recruitment, and experimental environment that are not controlled for. (ii) There are no detailed subjective measures of engagement, boredom, or fatigue or any other outcome behavioral outcome measure of control over alpha power during the trials that would allow a fine-grained comparison of mental state between Sham and Passive groups. While I agree that the pattern is consistent with a strong contribution of time-on-task, I would recommend softening this language and explicitly acknowledging the limitations of using an independent passive sample and the lack of detailed subjective state measures.

Now acknowledged, see new lines 716-719 and 845-850.